# *EpiCare*: A Reinforcement Learning Benchmark for Dynamic Treatment Regimes

**Mason Hargrave**
Center for Studies in Physics and Biology
The Rockefeller University
New York, NY, USA
`mhargrave@rockefeller.edu`

**Alex Spaeth**
Dept. of Electrical and Computer Engineering
University of California, Santa Cruz
Santa Cruz, CA, USA
`atspaeth@ucsc.edu`

**Logan Grosenick**[*]
Dept. of Psychiatry and BMRI
Weill Cornell Medicine, Cornell University
New York, NY, USA
`log4002@med.cornell.edu`

## Abstract

Healthcare applications pose significant challenges to existing reinforcement learning (RL) methods due to implementation risks, limited data availability, short treatment episodes, sparse rewards, partial observations, and heterogeneous treatment effects. Despite significant interest in using RL to generate dynamic treatment regimes for longitudinal patient care scenarios, no standardized benchmark has yet been developed. To fill this need we introduce *Episodes of Care* (*EpiCare*), a benchmark designed to mimic the challenges associated with applying RL to longitudinal healthcare settings. We leverage this benchmark to test five state-of-the-art offline RL models as well as five common off-policy evaluation (OPE) techniques. Our results suggest that while offline RL may be capable of improving upon existing standards of care given sufficient data, its applicability does not appear to extend to the moderate to low data regimes typical of current healthcare settings. Additionally, we demonstrate that several OPE techniques standard in the the medical RL literature fail to perform adequately on our benchmark. These results suggest that the performance of RL models in dynamic treatment regimes may be difficult to meaningfully evaluate using current OPE methods, indicating that RL for this application domain may still be in its early stages. We hope that these results along with the benchmark will facilitate better comparison of existing methods and inspire further research into techniques that increase the practical applicability of medical RL.

## 1 Introduction

Most human diseases evolve over time, many with trajectories that can be influenced by the right treatment [1]. Dynamic treatment regimes (DTRs) are adaptive medical policies which define a set of decision rules to determine the treatment to apply to a patient given the patient's medical history, including past treatments and observations [2]. Although latent biology drives disease progression, physicians lack direct access to the true biological state of any given patient and instead must rely on indirect and often partial clinical observations that correlate with this hidden state [1, 3, 4, 5, 6].

---

[*]To whom correspondence should be addressed.

38th Conference on Neural Information Processing Systems (NeurIPS 2024) Track on Datasets and Benchmarks.

Spurred by previous work applying reinforcement learning (RL) to other types of medical problems [7, 8], numerous authors have expressed interest in using offline RL to generate DTRs, especially in the case of longitudinal patient care with multi-treatment selection [9, 10, 11].

Medical RL models are faced with a chicken-and-egg problem: the RL models cannot be deployed until they are evaluated for safety, and cannot be directly evaluated except by being deployed. To address this, indirect pre-deployment validation methods are commonly used to evaluate the real-world readiness of various RL techniques. This pre-deployment validation can be approached in three ways. First, models can be trained on historical data, and their performance predicted via off-policy evaluation (OPE). Second, models can limit themselves to directly mimicking the behavior policy under which the historical data was collected, a process known as behavior cloning (BC) [12], which can avoid some issues with OPE by restricting the RL model's behavioral repertoire [13]. Finally, RL models can be trained on a simulated environment designed to capture the challenges expected in the real-world environment of interest. This simulation approach enables direct evaluation of RL policies on the simulated environment without ethical concerns. This approach also makes it possible to compare OPE performance predictions against the actual online performance of RL policies, providing a performance benchmark for OPE methods themselves. Despite the distinct advantages of the simulation-based approach, to date no such simulated environments have been developed for longitudinal healthcare applications — instead, most previous work has focused on simulating the effect of controlling individual drug dosages over short periods of time (See Section 2).

In this paper we introduce *Episodes of Care* (*EpiCare*), the first benchmark for RL in longitudinal patient care. We compare the performance of five state-of-the-art offline RL models on our benchmark. Additionally, we evaluate five common OPE methods to determine whether they reliably predict the performance of RL models when trained on *EpiCare's* simulated clinical trial data. Our findings indicate that these OPE methods cannot be trusted to accurately predict RL performance in longitudinal medical scenarios, calling into question their use for benchmarking RL performance in real-world clinical applications.

Key design considerations include:

**Realistic Difficulty.** *EpiCare* presents significant challenges for existing RL methods, including short episodes with varied initial conditions, unknown transition dynamics, and observation distributions that overlap between multiple distinct hidden states. Our benchmark also includes healthcare-specific challenges such as heterogeneous treatment effects (HTEs) and adverse events [14]. As we are chiefly interested in offline RL, we generate our off-policy datasets by way of simulated clinical trials which emulate the real-world collection of clinical data. While the challenges present in *EpiCare* are germane to the field of healthcare, *EpiCare* is designed as a benchmark capable of representing a class of medically inspired problems rather than a disease-specific simulation.

**Patient Safety.** One of the most important considerations in deployment of any new DTR is that it should not reduce patient safety relative to the existing standard of care (SoC). Therefore, in addition to mean returns, we also measure patient welfare statistics such as the adverse event rate and mean time to remission. For comparison, we model the SoC via a policy designed to emulate performance of a hypothetical clinician following best practice but without access to the latent disease states.

**Reproducibility and Configurability.** As an open source tool available on GitHub and conforming to OpenAI Gym standards [15], *EpiCare* aims to encourage the reproducibility and comparability of results critical to advancing the field of medical RL. The environment's configurability ensures that researchers can simulate a wide array of procedurally generated disease treatment scenarios of variable difficulty. We would like to stress that no such longitudinal medical treatment simulation environments exist and thus our work represents a first-in-class example of such a benchmark.

**Standardized Benchmarks.** While configurability is useful, having a standard benchmark is also important. To this end we have chosen some specific environment hyperparameters in close collaboration with medical professionals which reflect the realities of longitudinal patient treatment scenarios. As online RL has historically been too risky for most medical contexts [16], we focused our benchmarking efforts on offline RL methods, as well as off-policy evaluation (OPE).

## 2 Related Work

Reviews on both offline RL [17, 16], and medical RL [18] comprehensively cover a large scope of related work. An enormous fraction of the offline RL literature cites healthcare as a core motivation [19, 20, 21, 22], but evaluations typically use standard RL benchmarks that are unrelated to medicine [15, 23, 24]. This highlights a significant need for a healthcare-oriented RL benchmark like *EpiCare*.

A central problem in RL-generated DTRs is that of validating their real-world performance [25, 26, 27]. RL-generated DTRs are typically evaluated online; the DTR is applied to an environment for some number of episodes and the rewards are reported. In medical RL however, online evaluation is too risky prior to employing alternate initial validation strategies [16]. Instead, DTRs are evaluated by either off-policy evaluation (OPE) or via simulation, each having advantages and drawbacks.

**Off-Policy Evaluation on Real-World Data.** OPE is a class of techniques for predicting real-world performance of a policy by way of historical data [28, 29]. However, OPE is plagued by high data overheads and significant variance in the predicted performance [30]. Consequently, it has been claimed that most available medical datasets are not large enough for OPE [31]. Despite these challenges, numerous exciting RL contributions have emerged in the medical context, not only for discrete treatment selection in longitudinal patient care [9, 10, 11], but also for problems including propofol infusion control during surgery [32], mechanical ventilation for intensive care [33], sepsis treatment [34, 35, 36, 37], and chemotherapy [38]. Due to the widespread use of OPE to evaluate RL models trained on real-world data, much of the previous research on medical RL hinges on the quality of OPE methods themselves. Short of the ethically dubious proposition of deploying RL models directly on patient populations, simulated patient care models are the only other available pathway to validating OPE techniques. *EpiCare* represents such a benchmark and provides an unambiguous evaluation of OPE efficacy in longitudinal patient care scenarios.

**Simulated Environments.** In contrast to OPE, simulation-based methods evaluate the performance of RL algorithms on domain-specific pathogensis models. Most simulated environments in the medical RL literature are chiefly concerned with the continuous control of drug dosages. For example, an HIV drug dosage model [39] has been used by a number of researchers as a test bed for various RL techniques [40, 41, 42, 43]. Similarly, researchers have simulated blood glucose control for diabetes [44, 45, 46], anti-seizure medications for epilepsy [47], and levidopa dosage for Parkinson's disease [48]. Despite this focus on continuous control, it is common for clinicians to model disease progression dynamics as a set of discrete states which evolve over time (Figure 1a). While continuous models of medical scenarios like propofol infusion can be modulated to represent well-understood HTEs (especially those arising from known risk factors), we are not aware of any simulation which uses a discrete hidden state model to represent cryptic disease states. More broadly, there are no existing RL environments for longitudinal healthcare applications. This is despite the wealth of literature focused on the challenge of developing longitudinal treatment protocols for conditions specifically characterized by HTEs, such as acute respiratory effect syndrome [49], atrial fibrillation [50], osteoarthritis [51], and borderline personality disorder [52]. In this way *EpiCare* fills a critical gap in the existing medical RL literature.

## 3 Environment

*EpiCare* represents longitudinal patient care scenarios by modeling disease progression and treatment response over time (Figure 1b) using a Partially Observable Markov Decision Process (POMDP) framework (Figure 1c). The environment contains a state space representing various disease states including remission and adverse events, an observation space capturing clinical indicators (symptoms), and an action space representing the set of available therapeutic interventions. The probabilistic state transition dynamics are influenced by both the current state and selected treatment, while observations are emitted based on state-specific symptom distributions and modified by treatment effects. The reward function of *EpiCare* aligns with medical objectives to account for symptom management, treatment costs, and achieving remission. Each episode begins with a patient initialized in a random initial state, and the goal is to manage the patient's symptoms effectively through a sequence of treatment decisions until remission is achieved or the episode ends. *EpiCare* is highly configurable, allowing researchers to simulate a wide range of disease dynamics and treatment scenarios, providing a comprehensive benchmark for evaluating RL methods in longitudinal medical contexts. For the full modeling details of *EpiCare*, see Appendix A.

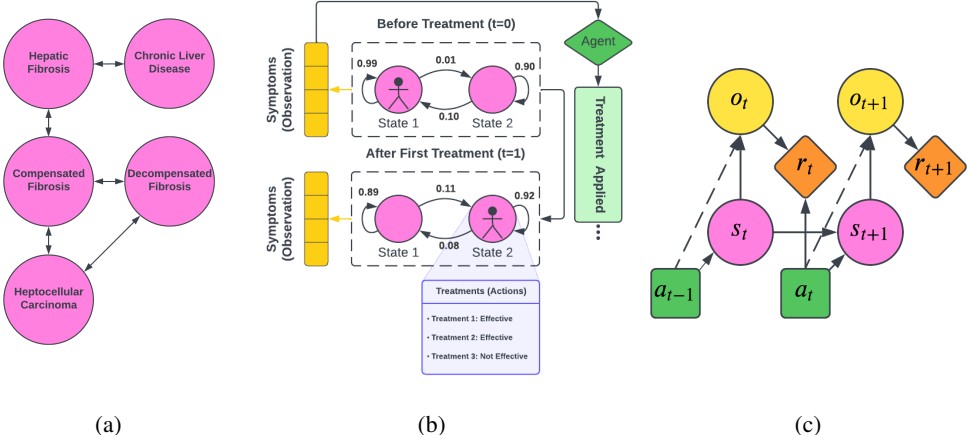

(a)                (b)                (c)

Figure 1: (a) A simple real-world example of the state transition graph for liver disease [53]. (b) A diagram representing a simple two-state disease. Inside the dashed boxes is a Markov model representing disease states. For each disease state, there exists a set of treatments which if applied may lead to remission (as indicated by the blue table), as well as a distribution of symptom severities. At the beginning of each episode, a patient is initialized in one of the disease states, and an initial observation of that patient's symptoms is collected. An agent then uses that observation to select a treatment to apply, which affects the transition probabilities out of the current state. This process continues until remission is achieved or a maximum number of timesteps is reached. (c) A graphical model of a POMDP complete with observations, rewards, states, and actions. The dashed lines from actions to observations indicate that in *EpiCare*, actions can directly affect observations.

An important feature of this model is that all of the POMDP parameters are generated pseudorandomly according to an "environment seed", which is separate from the random seed controlling the stochastic transitions within an episode. As a result, *EpiCare* defines a class of related environments indexed by the environment seed. The performance of an RL method should be evaluated across multiple environments in order to assess its generalizability. In this paper, we report the performance of each algorithm on eight different environment instantiations.

## 4 Policies

*EpiCare* includes three non-RL policies which serve two different purposes. First, they can be used to generate the datasets from which we train our offline RL algorithms of interest. When used in this way, the policies are referred to as "behavior policies". Second, they can be used as performance baselines against which to compare the performance of our RL models. When used in this way, the policies are referred to as "baseline policies".

These policies are not trained from data; instead, their behavior is computed directly from the parameters of the POMDP. These policies simulate medical decision-making (1) with complete state and state-specific treatment response knowledge (Oracle Policy), (2) without state or state-specific treatment response knowledge (SoC), and (3) using a popular real-world approach (SMART) for clinical trial randomization [54, 55]. For policies without state knowledge, it is possible to mis-estimate the efficacy of treatments, leading to worse performance compared to situations where states are identifiable (see Section 4.2). Overall performance of these policies is compared in Appendix B.3.

### 4.1 Oracle Policy (OP)

The oracle policy (OP) provides direct access to the hidden state, and at each timestep chooses the action which greedily maximizes the instantaneous expected reward given that state. This policy is not fully optimal, as it does not take into account multi-step treatment strategies (e.g. biasing

transition probabilities towards a disease state that would be easier to treat on the next step).[2] Still, the OP operates with significant advantage and can thus be used to establish a reasonable floor on best-case DTR performance.

## 4.2 Standard of Care (SoC)

The SoC policy aims to provide a facsimile of real clinician performance. Because our treatment scenarios are procedurally generated, however, there is no such thing as a real-world SoC to compare against. Therefore, we have made some assumptions about what such an SoC would look like. Because we are focused on the challenges associated with generating DTRs in scenarios with cryptic latent disease states and HTEs, we assume our idealized clinician does not have a way to estimate latent state or state-specific treatment effects. Instead, their knowledge of the medical literature and best practices is modeled by use of the ground truth expected reward of each action without hidden state information. Our SoC clinician also assumes that the reward distribution during each episode is non-stationary and patient-dependent. Thus each individual episode is a non-stationary multi-armed bandit with some known prior information, which we address using the common technique of an exponentially recency-weighted value estimate [56] that resets to the stationary expected reward at the beginning of each episode.[3] For implementation details, see Appendix B.1.

Given the importance of safety as a performance metric, a meaningful baseline policy must be able to take adverse events into account when selecting actions. Since adverse events occur when symptoms reach extreme values, our SoC policy simply avoids prescribing treatments which would worsen any symptom that is currently above a given threshold. The result is a greedy policy that simulates a plausible clinical SoC in the face of incomplete information about disease state.[4] This provides a conservative benchmark against which the performance of RL algorithms can be assessed.

## 4.3 Sequential Multiple Assignment Randomized Trial (SMART)

The SMART policy models treatment selection for a simulated sequential multiple assignment randomized trial (SMART) [54, 55]. This widely-used clinical trial strategy randomizes patients across multiple treatment arms. The policy adheres to a weighted random selection process where each treatment's likelihood of selection is based on its expected reward (for details, see Appendix B.2). This weighted sampling approach is inspired by Thompson sampling, a simple but effective heuristic approach for balancing exploration and exploitation [58]. The SMART policy allows us to generate synthetic clinical trial data which can be used to provide our RL approaches with relevant and realistic off-policy datasets.

## 5 Results

We employed *EpiCare* to benchmark five recent, high-impact offline RL methods: AWAC [59], EDAC [60], TD3+BC [61], IQL [62], and CQL [63]. Our implementations of these models are derivative of the CORL library [64]. Most of these models are usable for discrete control simply by optimizing the logits of a one-hot-encoded action output, but for TD3+BC and EDAC, it is necessary to propagate gradients through the chosen action; to convert these implementations for the discrete control case, we used Gumbel-Softmax reparameterization [65, 66]. Additionally, we benchmarked two simpler methods as baselines: behavior cloning (BC), and a deep $Q$ network (DQN) [67]. The input to each model consisted of not only the current symptoms, but also the entire observation history of the current episode as well as the last action selected. Hyperparameters were derived from sweeps carried out on each model according to ranges established in the literature (Appendix C.3). A diagram detailing the benchmarking process can be found in Figure 2.

---

[2]Greediness works well in the context of *EpiCare* because the environment is designed such that for every state there exists at least one effective treatment. If this were not the case, OP could fall into a local optima where it would only manage symptoms.

[3]Here the "arms" of the bandit are the possible actions/treatments at each timestep.

[4]Optimal multi-armed bandit policies must sometimes take actions which do not maximize instantaneous reward in order to gain information, due to the explore-exploit trade-off [57]. However, in the context of medicine, where the adage "first, do no harm" applies, exploration is prohibited, as exploring would be tantamount to enrolling a patient in a clinical trial without consent. As such, explore-exploit-based optimal solutions are not available to our idealized clinicians, resulting in the need for a greedy policy when modeling their behavior.

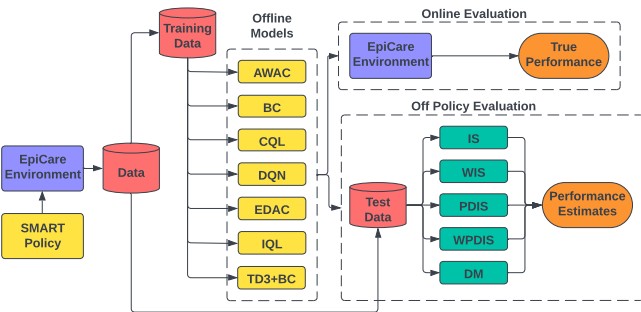

Figure 2: A diagram of the benchmarking process. The SMART policy was used to generate a synthetic clinical trial dataset from our environment. Once trained, the offline RL methods were evaluated both online and by way of OPE.

## 5.1 Online Evaluation

We assessed the performance of our chosen RL methods across variations of the environment by generating a dataset of $2^{17} = 131,072$ episodes from each of 8 different environment seeds collected under the SMART policy defined in Section 4.3.[5] Because the underlying POMDP is generated from the environment parameters, these datasets can be thought of as being drawn from sequentially randomized clinical trials of 8 unrelated diseases. These datasets, consisting of observation, action, and reward trajectories, were then used to train four replicates of each of our models of interest. Each trained model was evaluated on 1,000 episodes of online interactions. Online evaluations of OP and SoC are also reported, with SoC representing a lower bound on the acceptable performance of an RL algorithm. A policy which takes uniform random actions at all timesteps (Rand) was also assessed. The outcomes of this experiment can be found in Figure 3a and Table 4.

The RL methods we benchmarked fit broadly into two categories: value-based (CQL, DQN, and IQL) and actor-critic (TD3+BC, AWAC, and EDAC).[6] In our online evaluation metrics, all value-based methods outperformed all actor-critic methods for all metrics when averaged across environments. This makes some sense considering that our benchmarks used a relatively small action space of 16 treatments, and one of the main advantages of actor-critic methods is their ability to efficiently manage large (high-dimensional or continuous) action-spaces [68]. Indeed the advantage of value-based methods over actor-critic methods in discrete action spaces is well documented in other, non-medical domains [69]. Interestingly, TD3+BC, which is a hybrid between the actor-critic method TD3 and BC performs significantly better across the board than either method type in isolation (Figure 3a). We see similar relationships between model performance when it is quantified in terms of ability to achieve remission (Table 6). Overall, CQL, DQN, IQL, and to a lesser extent TD3+BC all outperform our SoC policy baseline, indicating that they learn to distinguish between the latent states. Of these, CQL has the best performance overall.

This advantage continues to a lesser degree in terms of adverse event rates, which we use to evaluate the safety of each RL method, i.e. the degree to which they avoid rare but negative consequences (Figure 3b). The adverse event rate metric also reveals that while DQN may achieve higher overall reward than IQL, IQL manages to trigger fewer adverse events. The safety disadvantage of DQN can be ascribed to its tendency to overestimate future rewards [70], a tendency which IQL and CQL are

---

[5]For a discussion on how training on SMART data compares to training on SoC data, see Section 5.4.

[6]BC, which we use as a performance baseline, is technically supervised learning as it does not use reward as a learning signal.

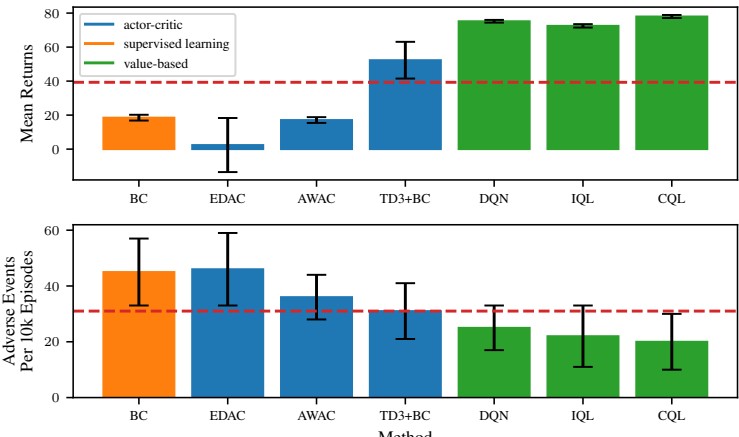

Figure 3: Performance evaluations in terms of (a) returns and (b) adverse event rates for all learning methods. These metrics are reported as their respective means across 4 replicates each of 8 structurally different *EpiCare* environments generated from environment seeds 1–8. The error bars represent the mean (across environments) of the standard deviation (across replicates). For comparison, the SoC baseline performance is shown as a horizontal dashed red line. See Tables 4 and 5 for full results.

both designed to correct against [62, 63]. Despite optimizing only for mean returns, CQL, IQL, and DQN all outperform SoC's heuristic approach in terms of adverse event rates, demonstrating that these methods have some ability to avoid actions which would lead to dangerous outcomes.

## 5.2 Data Restriction

For the results presented in Figure 3 we used $2^{17} = 131,072$ episodes worth of training data per environment. This quantity of simulated patients is well in excess of the typical size of clinical trials. Although clinical trials of individual treatments have in some cases had in the millions of patients, a more typical sample size would be in the hundreds, with the largest SMART trial including 2,876 patients [71, 72]. As such, it is important to evaluate how RL models perform in a restricted data regime. To test this, we trained the four top performing models from the initial evaluation with varying training set sizes to see how performance degrades as offline training data size decreases (Figure 4)[7]. We compare these to the OP, SoC, and Rand (random) policies, whose performance curves are constant horizontal lines because they are based on known environment parameters rather than learned from data.

DQN is the first model to beat SoC performance at 2,048 patients worth of data, very close to the size of the largest ever SMART clinical trial. In the low data regime, below 256 patients worth of data, DQN performance degrades below random. This is likely due to the fact that DQN has no mechanism by which to correct against reward overestimates, a problem that becomes more pronounced as data availability decreases. IQL in particular lags in terms of relative performance for a unique reason: the optimal number of IQL training steps varies as as function of data availability. For a full discussion of this peculiarity, see Appendix C.4. TD3+BC also exhibits an interesting phenomenon where the mean and variance of its returns decrease substantially near $N = 256$. We suspect that this may correspond to a double-descent-like effect, where the model (whose layers each have 256 neurons) moves out of the overparametrization regime, as was recently recorded in TD models [73]. We carry out the same analysis but with median remission rate instead of episode reward in Appendix C.5.

## 5.3 Off-policy Evaluation

A significant amount of previous work in medical RL is dependent on the belief that existing OPE methods behave as faithful estimators of the true real-world performance of a learned policy [13]. In the context of medicine however, the number of interactions with any given patient is usually

---

[7]The specific numerical results for the mean returns of these four models given a training set of only $2,048$ episodes can be found in Table 9

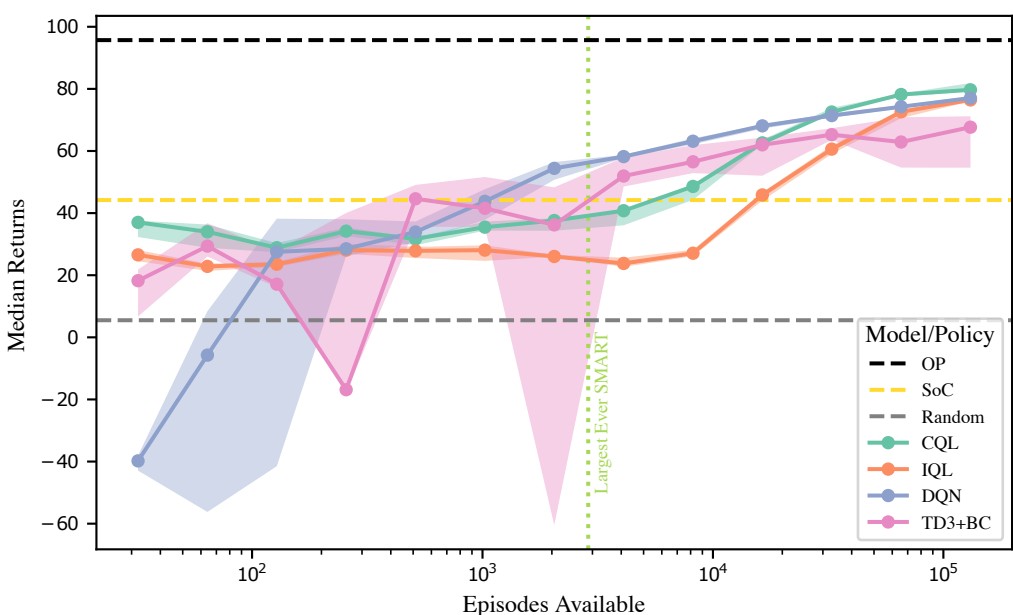

Figure 4: Median returns during the data restriction trials for four top performing RL models, compared to *EpiCare*'s three baseline policies (whose median per-episode performance is dictated only by the environment parameters and not by data availability).

Table 1: RMSE between the OPE estimates and the true online returns evaluated on 1,000 episodes for each combination of OPE method and RL model, across 8 seeds with 4 replicates. A plot of these results can be found in Appendix Figure 13.

|       | EDAC | AWAC | BC   | TD3+BC | IQL  | DQN  | CQL   |
|-------|------|------|------|--------|------|------|-------|
| IS    | 32.7 | 4.3  | 2.3  | 81.0   | 37.1 | 35.4 | 37.7  |
| WIS   | 61.4 | 4.3  | 2.3  | 36.2   | 35.0 | 10.9 | 10.7  |
| PDIS  | 35.6 | 4.4  | 2.3  | 112.8  | 38.3 | 57.9 | 56.0  |
| WPDIS | 36.7 | 8.4  | 6.7  | 46.3   | 30.7 | 44.4 | 46.9  |
| DM    | 23.0 | 11.8 | 12.3 | 50.6   | 46.9 | 93.5 | 106.4 |

quite small compared to existing RL benchmarks, a regime which is out of scope for existing OPE benchmarks [29, 74]. Therefore *EpiCare*, which incorporates unique challenges associated with healthcare including short episode lengths, can provide us with an optimistic picture of how well OPE is likely to work in the clinical setting. To this end we implemented five common OPE methods: IS, WIS, PDIS, WPDIS [75], and a simple direct method (DM) [76] based on a regression model of returns at each timestep. These methods were chosen based on their prevalence in the medical RL literature [9, 11, 34, 37]. In order to evaluate these OPE methods, we took the final model checkpoints for all RL models and conducted OPE on a 131,072 episode withheld test set. For the four importance sampling methods (IS, WIS, PDIS, and WPDIS), we used the mean value of the estimator across 8 bootstrap resamples of the test set. On the other hand, since DM is based on a trained model, instead of bootstrapping, we simply trained 4 replicates. Root-mean-square error (RMSE) between OPE estimates and online evaluations of each checkpointed model was then used to assess the degree to which the OPE estimates were indicative of the actual online evaluation results (Table 1).

We find that OPE estimates of online performance on *EpiCare* are poor overall, in line with the high reported variance of these estimators [30]. Furthermore, another key limitation of OPE is that its accuracy is dependent on an *effective* sample size, which can become orders of magnitude smaller than the number of data points available when the policy being tested differs significantly from the behavior policy [13], a fact which has been used to argue that RL in healthcare should be limited to behavior cloning policies [31]. Indeed, our OPE methods performed reasonably only on AWAC and BC, the two policies most likely to select the same action as the behavior policy (Appendix D).

Table 2: Mean return comparison between policies trained on SMART data vs. policies trained on SoC data. Mean (standard deviation) across 4 replicates on *EpiCare* environment 1. Only CQL and TD3+BC (italicized) outperform SoC when trained on SoC data.

| | EDAC | AWAC | BC | TD3+BC | IQL | DQN | CQL |
|---|---|---|---|---|---|---|---|
| SMART | 7.2(17.5) | 30.1(1.8) | 24.4(1.0) | 71.3(5.2) | 76.5(0.7) | 77.0(1.6) | 79.4(0.7) |
| SoC | -26.8(19.9) | 40.7(1.6) | 41.5(0.9) | *49.5(1.8)* | 42.6(0.9) | -59.6(7.0) | *54.3(0.4)* |
| Online | -3.2(0.8) | 64.4(0.7) | 66.7(0.8) | 9.6(1.0) | 68.7(0.7) | -32.6(0.7) | 63.9(0.7) |

## 5.4 Effect of Training Data

The quality of training data significantly affects offline RL performance. We evaluated this by training models on *EpiCare* data generated by three policies: the SoC policy (expert clinician behavior), the SMART policy (clinical trial simulation), and an online-trained DQN policy with the same hyperparameters as above (but with $2^{19}$ episodes) which we refer to simply as Online.

Results in Table 2 show that most models trained on SMART data outperform those trained on SoC or Online data, likely due to SMART's increased state space exploration through randomization. While SoC and Online policies are more effective for individual patients, their exploitative nature limits the diversity of training data. BC and AWAC by contrast, which both aim to replicate training data behavior, show improved performance when trained on higher-performing policies (Online > SoC > SMART), benefiting from the consistent, expert-driven behavior in SoC data and the patient-optimized decisions in the Online data.

These findings emphasize that more exploratory datasets may outperform expert-driven data for training robust RL policies in DTR healthcare, despite the latter's apparent advantages. Furthermore, methods like DQN, while effective with exploratory data, degrade significantly with less exploratory data likely due to overoptimism in unobserved contexts [63, 62].

## 6 Limitations

**Generalizability to Real-World Clinical Scenarios.** *EpiCare*, while sophisticated, clearly cannot capture all of the complexities of the clinic. We caution against attempting to use *EpiCare* as a model of any one particular disease without appropriate domain expertise both in terms of the disease of interest and the modeling details of the environment. The results of any disease-specific benchmarking should be audited independently by experts and ethicists for bias prior to deployment.

**Dependence on Simulation Parameters.** The performance of RL and OPE methods in *EpiCare* is influenced by the environment's parameters. Variations in parameters, such as the number of disease states or the connectivity of the states, could impact the relevance of our findings to specific contexts. In particular, we report results for 8 distinct *EpiCare* disease environments generated randomly from the same parameters. This leaves open the possibility that other parameters could yield more consistent OPE performance or faster RL convergence. However, we expect that real longitudinal medical care applications represent a greater challenge to existing methods than *EpiCare* such that our results act as a ceiling on real-world performance of both offline RL and OPE.

**OPE Methods.** Though we test a comprehensive list of the most common OPE methods in the medical RL literature, our list is not exhaustive. Still, we expect that our list is representative and that the same limitations would likely apply to OPE methods not included.

## 7 Guidance on Usage and Interpretation for Researchers

First and foremost, *EpiCare* is designed as a standalone medically inspired benchmark for RL and OPE methods. Any RL or OPE methods that cite longitudinal care as a motivating use-case should leverage *EpiCare* to validate the efficacy of the method in longitudinal healthcare contexts. To accomplish this, offline RL algorithms should be trained on the provided offline training datasets as generated by the SMART behavior policy, while online RL algorithms should simply train until

convergence on *EpiCare* itself.[8] RL algorithms that surpass the SoC baseline in performance are demonstrating clear evidence for the ability to distinguish between hidden states and associate effective treatments. Furthermore, RL algorithms with lower adverse event rates than the SoC baseline are in so doing demonstrating the ability to identify state and select safe treatments.

Given the high-degree of configurability in *EpiCare*, it may be tempting to set or fit the parameters of *EpiCare* to match some medical dataset or model some specific disease for sim-to-real applications. We caution against using *EpiCare* in this way as the configurable parameters are predominantly related to the random generation of different ensembles of fictitious disease environments. In this way *EpiCare* parameters are used to define a set of medically-inspired problems for RL to solve, indexed by environment seeds, rather than a single disease. Anyone interested in modeling a specific disease would likely be better off designing a more detailed simulation of a disease of interest including any disease-specific challenges not well-represented in the *EpiCare* benchmark.

A better way to use *EpiCare* in the context of applied medical RL research would be to set the benchmark parameters to ranges which are relevant to the disease of interest by asking questions like "How many unique treatments exist for my disease?", "What is the cure-rate for each treatment and how do they vary?", "How distinguishable are the hidden states believed to be and how many are there?", "How many symptoms or clinical measurements are associated with the disease?" and "How many time points do we typically have per patient?". These questions should guide the setting of the *EpiCare* parameters and allow researchers to titrate the challenges represented by *EpiCare*. Setting the parameters in this way should provide researchers with a *rough* estimate of the amount of data that would be necessary for any given RL method to be effective for a given medical use-case,[9] though we still caution as above that researchers may need to go beyond simply setting parameters to incorporate any disease-specific phenomena which are not well-accounted for with *EpiCare*.

## 8    Conclusion

Here we have introduced *EpiCare*, a comprehensive Python library designed to benchmark reinforcement learning (RL) methods in the context of medical treatment. We hope this work represents a significant stride towards benchmarking and realizing the practical application of RL in healthcare.

Our results demonstrate that existing OPE methods fail to provide reliable performance estimates even in our simplified model of clinical settings (inherently easier than real-world scenarios). This suggests that these methods are even less likely to succeed in the noisy and complex environments of actual clinical practice. The poor performance of OPE methods in our study calls into question the practical validity of much existing research that relies on these techniques for evaluating RL in clinical contexts. If OPE cannot reliably estimate model performance in *EpiCare*, the utility of OPE in more complex real-world scenarios is dubious—especially given that OPE depends on large data availability [13], and our simulated trials were orders of magnitude larger than standard and SMART clinical trials. Additionally, we show that while some RL methods can outperform our SoC baseline in terms of both efficacy and safety given sufficient data, this advantage disappears in the data-restricted regime typical of real clinical settings. Additionally, the superior performance of value-based methods over actor-critic approaches demonstrates the importance of method selection for medical applications. Finally, we show that of the value-based methods, both CQL and IQL have advantages over DQN with regards to safety (by way of lower adverse event rates) and with regards to learning from low-entropy training data as collected by highly exploitative behavior policies.

The medical community's increasing interest in RL-based dynamic treatment regimes demands rigorous evaluation methods. *EpiCare* addresses this need by providing a first-in-class benchmark that captures the key challenges of longitudinal healthcare settings. By enabling the systematic comparison of RL methods and evaluation techniques, we hope this work will facilitate more reliable assessment of RL's readiness for clinical applications and inspire new approaches better suited to the unique demands of healthcare settings.

---

[8]If training online, we recommend using the default environment parameters to maximize comparability.

[9]This estimate will vary between environment seeds, as *EpiCare*'s various complications can interact with each other in relatively unpredictable ways, affecting the difficulty of the environment. Any disease-specific benchmarking should consider multiple seeds with the same parameters, as in the present work.

## Acknowledgments and Disclosure of Funding

Thanks to Dr. Immanuel Elbau, Professor Marcelo Magnasco, and Alexander Epstein for helpful discussions, and to Professor Marcelo Magnasco for GPU computer resources. MH was supported in part by an NSF NGRFP. LG is supported by NIH R01MH131534, R01MH118388, New Venture Fund 202423, a Whitehall Foundation grant (WF 2021-08-089), a Cornell Center for Pandemic Prevention Research seed grant, and an A2 Collective pilot grant (PennAITech, NIA).

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

# A  Environment Continued

*EpiCare* models longitudinal care as a POMDP [77] whose state space is denoted by $\mathcal{S} = \{s_r, s_a, s_1, s_2, \ldots, s_{n_s}\}$ and consists of $n_s$ distinct disease states, together with two terminal states $s_r$ and $s_a$ representing termination of a treatment episode due to either remission or an adverse event respectively. The action space $\mathcal{A} = \{a_1, a_2, \ldots, a_{n_a}\}$ is a discrete set of $n_a$ available treatments. Finally, the observation space $\mathcal{O}$ is an abstract representation of clinical indicators that could potentially be measured at every timestep in an episode. Observations could be any combination of measurements taken by a clinician, but for simplicity we will refer to them as just "symptoms". Observations are normalized so that 0 signifies the absence of symptoms, and 1 signifies the most severe symptom presentation possible. We assume there are $d_o$ separate symptoms, so that $\mathcal{O} = [0, 1]^{d_o}$. Table 3 summarizes all parameters available for configuring the environment.

## A.1  State Transitions and Remission

Disease progression is characterized by a transition function $T(s'|s, a)$ which gives the probability of transitioning to state $s'$ at step $t + 1$ given both the state $s$ and action $a$ at time $t$ under the common assumption of time-homogeneity [78, 79].

Remission can occur from any disease state $s_i$ with a treatment-dependent probability $T(s_r|s_i, a)$. Adverse events, i.e. transitions into $s_a$, are modeled based on the observations — if any symptom exceeds a threshold $o_a^*$, the state transitions directly to the terminal state $s_a$.[10]

If a given action does not result in remission or an adverse event, the environment transitions to a different disease state based on an autonomous transition matrix $\mathbf{T}$ affected by an action-dependent modulation vector $\mathbf{m}_a \in \mathbb{R}_{>0}^{n_s}$ intended to capture the effect of treatments on state transitions. The probabilities of transitions between disease states are calculated by multiplying the transition probabilities by $\mathbf{m}_a$, then renormalizing such that the sum of state transition probabilities is equal to the probability that the state does not transition into remission or an adverse event, as follows:

$$T(s_j|s_i, a) = (1 - T(s_r|s_i, a) - T(s_a|s_i, a)) \frac{(\mathbf{m}_a)_j \mathbf{T}_{i,j}}{\sum_{k=1}^{n_s} (\mathbf{m}_a)_k \mathbf{T}_{i,k}}. \tag{1}$$

An important property of our environment is that the disease transition dynamics are sparse (see Figure 5), as in the liver disease example of Figure 1a [53]. The use of a multiplicative modulation $\mathbf{m}_a$ allows actions to affect the dynamics while preserving the sparsity of $\mathbf{T}$. We generate these dynamics from the environment seed according to Algorithm 1.

---

**Algorithm 1** Base Transition Matrix Generation

---

1: **initialize** $\mathbf{T} \leftarrow \mathbf{I}_{n_s}$
2: **for** $(i, j)$ in $\{(x, y) \mid 2 \leq x \leq n_s, 1 \leq y < x\}$ **do**
3:     **sample** $p \sim \mathcal{U}_{[0,1]}$
4:     **if** $p < p_c$ **then**
5:         **sample** $\mathbf{T}_{i,j} \sim \mathcal{U}_{I_{\mathbf{T}}}$
6:         **sample** $\mathbf{T}_{j,i} \sim \mathcal{U}_{I_{\mathbf{T}}}$
7:     **end if**
8: **end for**
9: **for** $i = 1$ to $n_s$ **do**
10:     **let** ROWSUM $:= \sum_{j=1}^{n_s} \mathbf{T}_{i,j}$
11:     **for** $j = 1$ to $n_s$ **do**
12:         $\mathbf{T}_{i,j} \leftarrow \mathbf{T}_{i,j}/$ROWSUM
13:     **end for**
14: **end for**

---

Algorithm 2 is then used to generate the values of $T(s_r|s, a)$ (arranged into a matrix $\mathbf{P}$), which guarantees that each state is treatable via at least one action. In the algorithm, $\mathcal{S}_d$ refers to the set of

---

[10]A true POMDP cannot include transitions which depend on observations, but this framing is much simpler notationally. In order to recover a proper POMDP, it would be necessary to truncate all observation distributions at $o_a^*$ and add a probability of transition to $s_a$ equal to the truncated area.

Table 3: Configurable parameters in *EpiCare* with default ranges and values. Parameters with default distributions are sampled and fixed at environment initiation based on a random seed passed to the environment. Parameters which have a non-empty index column indicate that the parameter in questions is sampled iid such that there exists a uniquely sampled value of the parameter for every element of the space.

| Environment Parameter | Symbol | Type | Indexed By | Default |
|---|---|---|---|---|
| Number of treatments | $n_a$ | Integer Value | - | 16 |
| Number of disease states | $n_s$ | Integer Value | - | 16 |
| Number of symptoms/indicators | $d_o$ | Integer Value | - | 8 |
| Maximum num. treatment courses | $v$ | Integer Value | - | 8 |
| Remission reward | $r_\mathrm{r}$ | Continuous Value | - | 64 |
| Adverse event penalty | $r_\mathrm{a}$ | Continuous Value | - | $-64$ |
| Adverse event threshold | $o_\mathrm{a}^*$ | Continuous Value | - | 0.999 |
| Symptom cost | $c_o$ | Continuous Value | - | $r_\mathrm{r}/(2vd_o)$ |
| State connection probability | $p_c$ | Continuous Value | - | $1/n_s$ |
| Num. diseases treatment cures | $n_{s|a}$ | Integer Value | $\mathcal{S}$ | $\sim \mathcal{U}_{\{1\ldots n_a/8\}}$ |
| Num. symptoms affected by treatment | $d_{o|a}$ | Integer Value | $\mathcal{A}$ | $\sim \mathcal{U}_{\{1\ldots d_o\}}$ |
| Cost of treatment | $c_a$ | Continuous Value | $\mathcal{A}$ | $\sim \mathcal{U}_{[1, r_\mathrm{r}/(2v)]}$ |
| Symptom modification vector | $\boldsymbol{\delta}_a$ | Continuous Vector | $\mathcal{A}$ | $\sim \mathcal{U}_{[-2,1.0]}^{d_o}$ |
| Transition modulation vector | $\mathbf{m}_a$ | Continuous Vector | $\mathcal{A}$ | $\sim \mathcal{U}_{[0.5,1.5]}^{n_s}$ |
| Symptom mean range | $I_\mu$ | Continuous Range | - | $[0, 2]$ |
| Symptom std. range | $I_\sigma$ | Continuous Range | - | $[1, 2]$ |
| Remission probability range | $I_\mathrm{r}$ | Continuous Range | - | $[0.8, 1.0]$ |
| Transition probability range | $I_\mathbf{T}$ | Continuous Range | - | $[0.01, 0.2]$ |

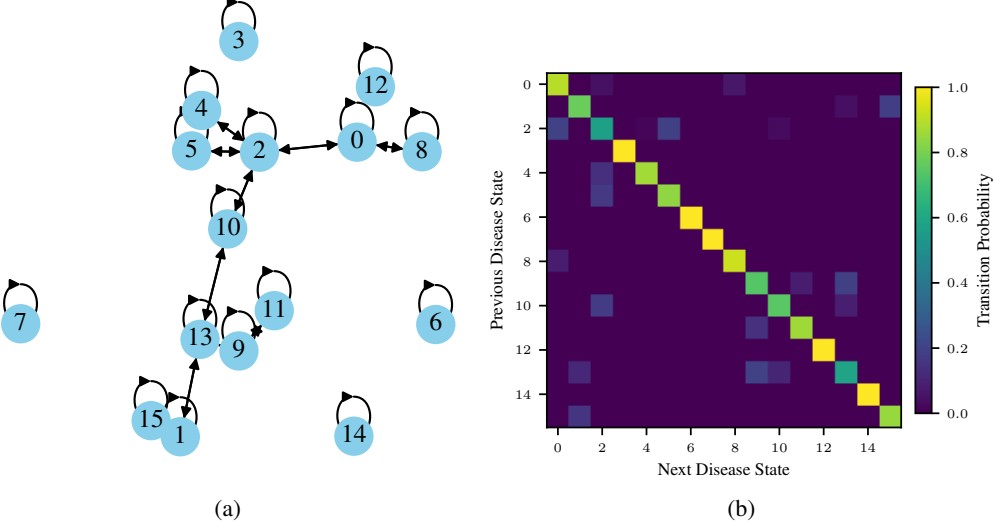

(a)  (b)

Figure 5: The (a) connectivity graph and (b) transition matrix $\mathbf{T}$ of the disease states generated by *EpiCare* for environment 1.

all disease states, i.e. every state other than remission and adverse events. This can be defined as:

$$\mathcal{S}_\mathrm{d} = \mathcal{S} \setminus \{\mathrm{s}_\mathrm{r}, \mathrm{s}_\mathrm{a}\} = \{s1, s2, \ldots, s_{n_r}\}.$$

---

**Algorithm 2** Generate Remission Probabilities for Each Action

---

1: **initialize** $n_a$ by $n_s$ zeros matrix $\mathbf{P}$
2: **set** REMAINING_STATES $\leftarrow \mathcal{S}_\mathrm{d}$
3: **for** $a$ in $\mathcal{A}$ **do**
4:     **sample** $n_{s|a} \sim \mathcal{U}_{\{1\ldots n_a/8\}}$
5:     SELECTED_STATES $\leftarrow$ **sample** $n_{s|a}$ states from $\mathcal{S}_\mathrm{d}$
6:     REMAINING_STATES $\leftarrow$ REMAINING_STATES $\setminus$ SELECTED_STATES
7:     **for** $s$ in SELECTED_STATES **do**
8:         **sample** $\mathbf{P}_{s,a} \sim \mathcal{U}_{I_r}$
9:     **end for**
10: **end for**
11: **for** $s$ in REMAINING_STATES **do**
12:     **sample** $a$ from $\mathcal{A}$
13:     **sample** $\mathbf{P}_{s,a} \sim \mathcal{U}_{I_r}$
14: **end for**

---

## A.2 Observations

Each disease state has an associated constellation of symptoms which could be confounded by a variety of factors, including fluctuation over time, measurement noise, finite measurement resolution, correlations between symptoms, and the effects of treatment. We model this by generating observations as

$$\mathbf{o} = \big[\operatorname{expit}(\tilde{\mathbf{o}} + \boldsymbol{\delta}_a)\big]_1, \tag{2}$$

where the symptoms $\tilde{\mathbf{o}}$ of the current state are chosen randomly at each timestep from a state-dependent distribution, then combined with a constant confounding vector $\boldsymbol{\delta}_a$ induced by the treatment. The sum is kept within the symptom range $[0, 1]$ by the sigmoidal function $\operatorname{expit} x = \frac{1}{2}\tanh\frac{x}{2} + \frac{1}{2}$, then quantized[11] to have only one digit past the decimal point in order to model finite-resolution effects. The underlying symptom vector $\tilde{\mathbf{o}}$ is drawn from a multivariate Gaussian distribution, which provides a first-order model of symptom interactions. This distribution has separate means $\boldsymbol{\mu}_s$ and covariance $\boldsymbol{\Sigma}_s$ in each nonterminal state $s$, which are generated according to Algorithm 3. The algorithm generates independent mean and standard deviation for $d_o$ observations, then uses a random orthonormal matrix to transform the distribution into coordinates where observations will be correlated.

---

**Algorithm 3** Generate Observation Parameters $\boldsymbol{\mu}$ and $\boldsymbol{\Sigma}$ for Each State

---

1: **for** $s$ in $\mathcal{S}_\mathrm{d}$ **do**
2:     **sample** $\boldsymbol{\mu}_s \sim \mathcal{U}_{I_\mu}^{d_o}$
3:     **sample** $\boldsymbol{\sigma}_s \sim \mathcal{U}_{I_\sigma}^{d_o}$
4:     $\boldsymbol{\sigma}_s \leftarrow \operatorname{sort}(\boldsymbol{\sigma}_s)$
5:     **sample** $A \sim \mathcal{N}(0,1)^{d_o \times d_o}$
6:     **let** $Q, R := \operatorname{QR}(A)$
7:     **let** $P := Q$
8:     **let** $\boldsymbol{\Sigma}_s := P \cdot \operatorname{diag}(\boldsymbol{\sigma}_s^2) \cdot P^T$
9: **end for**

---

The resulting observation model encapsulates state and treatment-specific effects on symptom observations, as well as several kinds of confounding. In general, the more distinguishable two states are on the basis of observations, the easier it becomes for reinforcement learning algorithms to generate an effective policy. Noise, treatment effects, and quantization all play key roles in determining state

---

[11]We use the notation $[\,\cdot\,]_1$ to denote rounding to one decimal place.

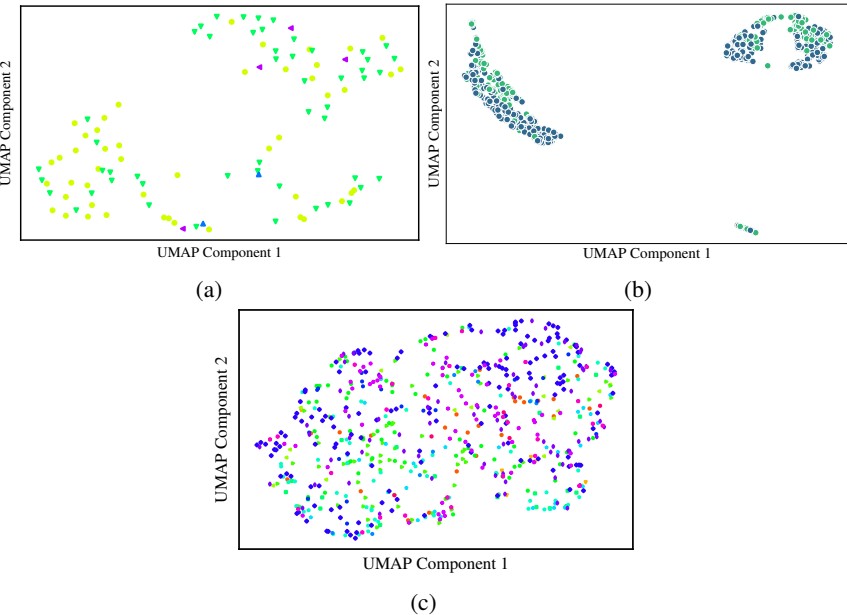

Figure 6: (a) UMAP embedding of clinical observations for $N = 105$ patients with depression in an fMRI dataset known to contain latent subtypes. Different markers represent the four unique subtypes previously calculated for this data subset. (b) UMAP embedding of various biometric features considered as predictors of diabetes risk for $N = 768$ patients. Green markers indicate patients later diagnosed with diabetes. (c) UMAP embedding of the observations of 100 episodes under the random policy in *EpiCare* environment 1, colored by ground truth hidden state.

separability. We tuned the default environment hyperparameters provided in *EpiCare* so that states would not be trivially separable (Figure 6c).

For an illustrative real-world example of disease states which are not trivially separable on the basis of observation, we performed a simple analysis of a functional magnetic resonance imaging (fMRI) dataset comprising resting state functional connectivity (RSFC) measurements as well as clinical rating scales for depression severity [80, 81] for $N = 105$ patients undergoing treatment for depression. Four biotypes for depression have been proposed on the basis of fMRI data [82], for which a machine learning classifier was recently developed [83]. We applied this biotyping procedure to each patient in the $N = 105$ subject dataset, then performed a UMAP projection of the clinical observations for each patient. Figure 6a compares these results to a UMAP embedding of observations from *EpiCare* environment 1 colored by ground truth hidden state, revealing comparable degrees of state ambiguity. The same UMAP projection process was carried out on clinical and demographic features from $N = 768$ patients with and without diabetes, with similar results as shown in Figure 6b [84].

### A.3 Reward Structure

Our reward function $R$ is designed to align the an RL agent's actions with the overarching goals of effective disease management, including minimizing symptoms, reducing treatment costs, and achieving remission. Consistent with the medical paradigm described above, we assume that states are inaccessible except indirectly through observations (outside of remission and adverse events, which we model using state-based rewards for simplicity). Thus our reward function $R(s, a, \mathbf{o}) : \mathcal{S} \times \mathcal{A} \times \mathcal{O} \to \mathbb{R}$ can be written as:

$$R(s, a, \mathbf{o}) = \begin{cases} r_{\mathrm{r}} & \text{if } s = \mathrm{s_r} \\ r_{\mathrm{a}} & \text{if } s = \mathrm{s_a} \\ -c_a - c_o \sum_{i=1}^{n_o} \mathbf{o}_i & \text{otherwise,} \end{cases} \quad (3)$$

with $r_{\mathrm{r}}$ being the reward for achieving remission, $r_{\mathrm{a}}$ the penalty associated with an adverse event (by default $r_{\mathrm{a}} = -r_{\mathrm{r}}$), and $c_o$ a scaling constant for costs associated with symptom-severity intended to

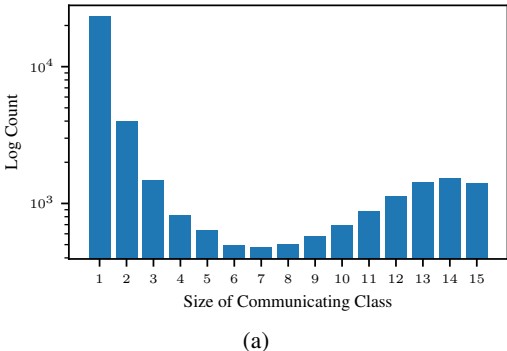
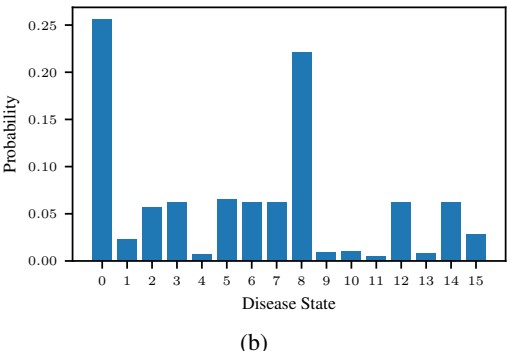

| (a) | (b) |

Figure 7: (a) The distribution of sizes of communicating classes across 10,000 different environment seeds. (b) The stationary distribution $\phi_0(s)$ across disease states for environment 1 of *EpiCare*.

penalize poor symptom management. The costs $c_a$ are intrinsic treatment-specific quantities used to represent clinical realities, such as financial burden, invasiveness, and general risk. We report rewards rescaled by $100/r_r$ so that the maximum achievable episode reward is 100 regardless of environment parameters.

### A.4  Initial State Distribution

At the beginning of each episode, an initial state $s_0$ is sampled from an initial state distribution $\phi_0$. All POMDP parameters remain constant, so that each episode represents a patient in the same population. A uniform distribution might seem an intuitive choice here, but does not respect the long-term state occupancy rates expected given our base transition matrix $\mathbf{T}$. Instead, we calculate an initial state distribution under the assumption that an initial uniform distribution has been allowed to evolve according to $\mathbf{T}$ for many timesteps without the influence of any treatment actions.

If $\mathbf{T}$ were irreducible and aperiodic, there would exist a unique stationary distribution $\phi$ over the states such that $\phi\mathbf{T} = \phi$, however $\mathbf{T}$ is generated such that it may not satisfy these conditions, and consequently $\phi$ may not be unique [85]. To address this, we can identify the communicating classes within $\mathbf{T}$, represented by subsets $C_1, C_2, \ldots, C_k$. This can be accomplished by a variant of Tarjan's algorithm implemented in *SciPy* [86]. Each class $C_i$ is a set of states that are mutually reachable; these can be thought of a distinct patient subtypes. The transition matrix restricted to each subtype, $\mathbf{T}|_{C_i}$, satisfies the criteria for the existence of a unique stationary distribution $\phi_{C_i}$ which can be found for all $C_i$ using a linear solver [87].

We then establish an initial distribution across these subtypes $\boldsymbol{\tau} = (\tau_1, \tau_2, \ldots, \tau_k)$ where each $\tau_i$ corresponds to the proportion of the population initialized in subtype $C_i$. Given $\boldsymbol{\tau}$, the global stationary state distribution $\phi$ is thus a positive linear combination of the stationary distributions of the subtypes: $\phi(\boldsymbol{\tau}) = \sum_{i=1}^{k} \tau_i \cdot \phi_{C_i}$. For our simulations, we take $\boldsymbol{\tau}_0 = (\frac{|C_1|}{n_s}, \frac{|C_2|}{n_s}, \ldots, \frac{|C_k|}{n_s})$ where $|C|$ is the number of states in $C$ (see Figure 7a). This allows us to choose a specific stationary distribution which we will also take to be our initial state distribution $\phi_0 = \phi(\boldsymbol{\tau}_0)$ (e.g. Figure 7b). There is no remission probability without treatment, so $\phi_0(s_r) = 0$.

Because our model is based on disjoint communicating classes of disease states, two patient sub-populations separated by a non-cryptic factor (e.g. young/old) could easily be represented as two separate patient subtypes which happen to have similar but not identical dynamics. It is definitionally impossible for a patient to transition between distinct communicating classes, and therefore *EpiCare* is sufficiently general to include HTEs caused by known factors.

### A.5  Default Environment Hyperparameters

*EpiCare* is highly configurable, though specific environment hyperparameters were chosen for the sake of our benchmarks. These default values can be seen in Table 3. The number of treatments $n_a$, disease states $n_s$, and symptoms $d_o$ were chosen in consultation with clinicians to reflect reasonable orders of magnitude encountered in real-world medical practice. Similarly, the maximum number of

treatment courses $v$ was chosen as a high but reasonable number of treatment courses a clinician may be able to attempt before the patient becomes non-adherent. The symptom and treatment costs $c_o$ and $c_a$ were set such that the worse-case treatment trajectory would achieve a negative reward equal in magnitude to the positive reward $r_r$ attributed to remission, assuming no adverse events occur. The adverse event penalty $r_a$ is set to the negative of the remission reward, so the true minimum episode reward is $-2r_r$.

The state connection probability $p_c$ was chosen because $1/n_s$ is the critical point in the phase transition of an Erdős-Rényi random graph to having a single giant component [88]. Depending on the environment seed, the result is typically a few large communicating classes of disease states, together with a few smaller components or isolated states (Figure 5a). If the goal were to have random graphs with a giant component, one could increase the connection probability, and if the goal were to increase the number of communicating classes, one could decrease the connection probability.

The range of the $d_{s|a}$, number of symptoms affected by a each treatment, was chosen to be maximally large for the sake of generality. On the other hand, the number of disease states each treatment could cure was chosen as to make treatments fairly state-specific (i.e. to ensure that some strategy was required to pick the correct treatment for each patient). The remission probability range was chosen to ensure that if the correct treatment were applied to a given state, it would be highly likely to be effective, doubling down on our interest in the state-specific treatments.

The symptom modification vector $\boldsymbol{\delta}_a$ was sampled such that treatments are more likely to positively affect symptoms than negatively affect them, while the transition modulation vector $\mathbf{m}_a$ was sampled as to affect but not dominate existing disease state transition dynamics. Finally, the transition probability range was tuned such that the typical episode would incur at least one transition.

We would like to emphasize that our design choices represent only one set of reasonable choices once could make, and other researchers may benefit from modifying our assumptions to benchmark RL methods for their specific use case. We have worked to keep our framework highly modular to allow easy incorporation of different distributions and extensions by future users of *EpiCare*.

## B  Baseline Policies Continued

### B.1  SoC Policy Details

This section presents the mathematical formulation of the SoC policy described in Section 4.2. Note that this depends on POMDP notation established in Appendix A. In the following, we will use the expression $q_*(s, a)$ to represent the instantaneous expected reward of the action $a$ in the state $s$, i.e. $q_*(s, a) = \mathbb{E}[R|s, a]$. The exponentially recency-weighted value estimate of an action $a$ at a timestep $t$ within an episode is denoted by $Q_t(a)$.

At the start of each episode (first interaction with each patient), the value estimate is reinitialized to the ground truth population expected instantaneous reward for the clinician's first action:

$$
\begin{aligned}
Q_0(a) &:= \mathbb{E}[R|a] = \mathbb{E}_{s \sim \boldsymbol{\phi}_0}[q_*(s, a)|a] \\
&= \sum_{i=1}^{n_s} q_*(s_i, a)\boldsymbol{\phi}_0(s_i),
\end{aligned}
\tag{4}
$$

where $n_s$ is the number of states and $\boldsymbol{\phi}_0(s)$ is the probability of a given patient being in state $s$ according to the initial state distribution. As the clinician continues interacting with the patient, this initial estimate undergoes updates according to:

$$
Q_{t+1}(a) = \begin{cases} Q_t(a_t) + \alpha[R_t - Q_t(a_t)] & \text{if } a = a_t \\ Q_t(a) & \text{otherwise,} \end{cases}
\tag{5}
$$

where $R_t$ is the reward received at timestep $t$ and $\alpha$ is a real value between 0 and 1.

Greedily maximizing the above reward estimate would define a state-agnostic policy which could act as a performance baseline. However, it does not take into account adverse effects, so we additionally prohibit the SoC policy from choosing actions which could worsen symptoms that are already high. Specifically, we introduce a threshold parameter $\kappa$ and define an observation-dependent set $\mathcal{A}_{\text{safe}}(\mathbf{o})$ of safe actions, i.e. the set of all treatments $a$ whose effect $(\boldsymbol{\delta}_a)_i$ on symptom $i$ is not positive for any

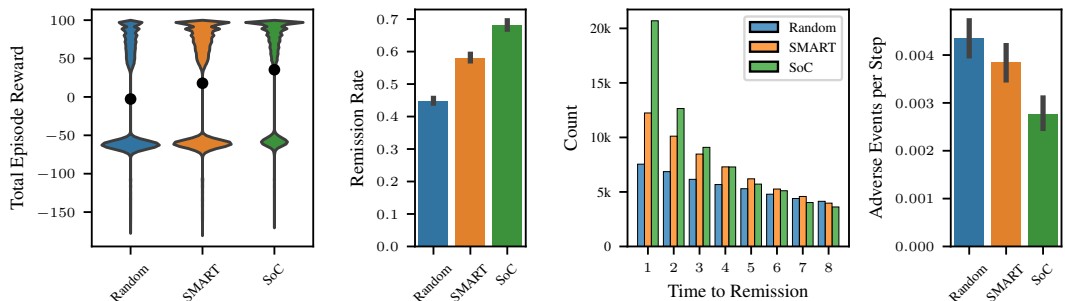

Figure 8: Comparison of various policies. Total episode reward (a), remission rate (b), time to remission (c), and adverse event rate (d) are all better for SoC than SMART and better for SMART than for random.

$i$ where $\mathbf{o}_i \geq 1 - \frac{\kappa}{2}$. The hyperparameters $\alpha$ and $\kappa$ were optimized for a combination of performance metrics as described in Appendix B.4.

## B.2 SMART Policy Details

This section presents the mathematical formulation of the SMART policy described in Section 4.3. In the following, we use the notation $q_*(a)$ to represent the stationary expected reward of the action $a$ across all states, i.e. the expected reward $\mathbb{E}_{s \sim \phi_0}[R|a]$ with $s$ distributed according to the stationary distribution $\phi_0$ described in Appendix A.4.

The treatment $a_t$ for a given step $t$ is determined by a weighted sample from $\mathcal{A}$. The weights $w_a$ of each action are defined so that the log probability of each action is proportional to its reward, but rescaled to ensure that the action with the highest reward estimate was a fixed ratio $\beta_a$ times more likely to be chosen than the action with the lowest reward estimate. We define the rescaled reward values $\hat{Q}_a$ for each action as:

$$\hat{Q}_a := \ln \beta_a \frac{q_*(a) - \max_a Q(a)}{\min_a Q(a) - \max_a Q(a)},$$

which then yields our weights: $w_a = e^{-\hat{Q}_a} / \sum_a e^{-\hat{Q}_a}$. In the current study, we use $\beta_a = 8$ as this provides a reasonable balance between exploration and exploitation.

## B.3 Performance Comparison

Figure 8 compares the performance of the SoC, SMART, and uniform random policies across 1000 episodes each for 100 distinct *EpiCare* environments using four different metrics: mean returns, remission rate, time to remission, and adverse event rate. The return (also called episode reward) is the total undiscounted reward for the episode, shown as a distribution across all episodes for all 100 environments. The bimodality in the violin plot is caused by the large remission reward leading to large difference in total reward between episodes where remission was achieved and for those which it was not achieved. There is also a long lower tail coming from infrequent but consequential adverse events.

Remission rate is the fraction of 1000 episodes in which remission is eventually reached, averaged across 100 distinct *EpiCare* environments, with error bars representing a 95% confidence interval.[12] Remission time is the number of actions taken before remission given that remission occurs, shown as a histogram for all episodes in which remission was eventually reached across all environments. Finally, the rate of adverse event occurrence is given as a bar graph in the same format as the remission rate. This is expressed as the average probability of an adverse event at each timestep in order to more directly measure safety: the absolute number of adverse events is indirectly decreased simply by increasing remission rates so that the patient spends less time in disease states.

---

[12]For this confidence interval we assume normality of errors.

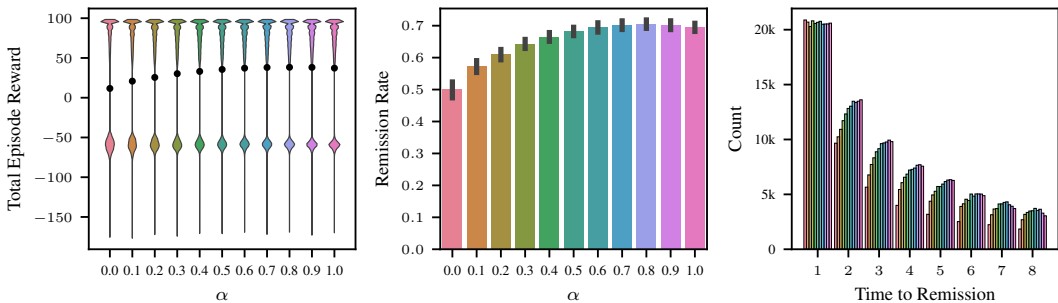

Figure 9: The $\alpha$ value of the SoC policy as it affects 1000 episodes across 100 different seeds. This does not significantly affect the adverse event rate, so that panel is omitted relative to Figure 8 for space.

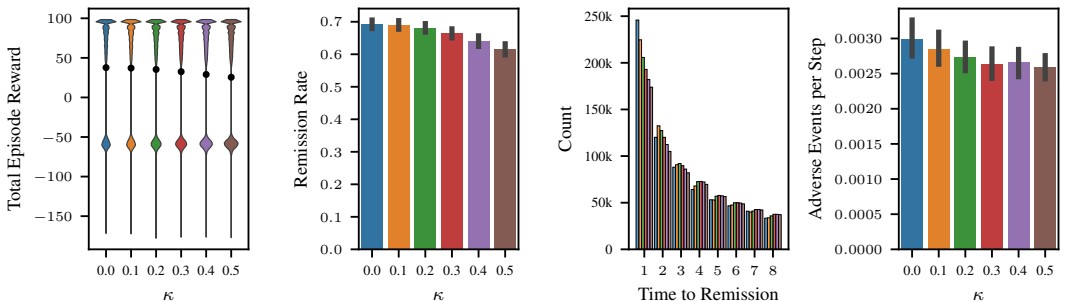

Figure 10: The $\kappa$ value of the SoC policy as it affects 10,000 episodes across 100 different seeds. The adverse event rate is expressed as the probability of an adverse event in each timestep, in an attempt to control for any variation in performance due to changing $\kappa$.

### B.4 Hyperparameters of SoC

We tuned the values of $\alpha$ and $\kappa$ used throughout our comparisons by empirically comparing the performance of various values across 100 different *EpiCare* environments. First we chose the value $\alpha = 0.8$ to maximize remission rate (Figure 9). We then tuned the value of the threshold $\kappa$ above which a symptom is considered potentially dangerous (so the SoC avoids treatments which increase that symptom) in exactly the same way (Figure 10), and chose a value of $\kappa = 0.2$ in order to decrease the risk of adverse effects as much as possible without significantly reducing mean outcomes. Since adverse events are relatively rare and the effect size is quite small, we evaluated this on 10 times more episodes than in other cases in order to get a more accurate estimate of the adverse event probability. The selection of $\kappa$ determines the degree of risk-averse behavior exhibited by the state-agnostic clinician modeled by the SoC policy.

### B.5 Switching Treatments is Evidence of Belief in States

To understand the effect of the value of $\alpha$ on the performance of the SoC policy, consider two simple extreme cases. One of these extreme cases is the one where the state does not depend on its previous values, i.e. every entry of the transition matrix $\mathbf{T}$ is taken to be $1/n_s$. In this maximum-entropy case, the knowledge that a treatment was ineffective at one timestep does not provide any information about the state of the patient at the next timestep, and therefore the best posterior estimate of the treatment's value is still equal to the prior, that is $Q_{t+1}(a) = Q_t(a)$ for all $t$. Since the patient's response is always drawn from the population distribution, the clinician should repeatedly apply the treatment maximizing $Q_0(a)$ at every timestep regardless of the patient's response.

On the other extreme, we can also imagine a simplification of the system where $\mathbf{T} = \mathbf{I}_{n_s}$. This is to say that patients, once initialized, do not deviate from their initial state. When a treatment does not lead to remission, the patient-specific posterior distribution of its value should decrease substantially, which is likely to lead to a new treatment being believed optimal for this patient so long

as treatments are somewhat state-specific. Thus the clinician will continuously try different treatments, but sometimes reapply previous treatments when others have been ruled out more conclusively.

Given a transition matrix $\mathbf{T}$ whose behavior is somewhere in between the maximum-entropy and fixed-state cases, we would also expect a state-agnostic clinician to switch between treatments in a similar way. Obviously clinicians commonly switch between treatments for a given disease population trying the efficacy of various treatments for a given patient in order to determine what works best for them. This demonstrates that clinicians believe in the existence of patient disease states, making treatment decisions based on the implicit state transition structure and treatment selectivity of the disease which they are treating.

Low values of $\alpha$ are ideal for situations close to our maximum entropy limit example or with low treatment selectivity, and high values of $\alpha$ are ideal for situations close to our stationary limit example or with high treatment selectivity. In essence this value controls the readiness the policy has to update its estimates of the reward.

## C    Training & Additional Results

### C.1    Online Results Continued

The full results containing a breakdown for baseline and model performance across all 8 environments produced by environment seeds 1-8 in terms of both mean returns and adverse event rates (as shown in Figure 3) can be found in Table 4 and Table 5.

We also measured the success of trained models by their probability of achieving remission (remission rate), and the mean length of episodes in which remission was achieved (remission time). This is intended to provide a more disease-focused metric that answers essentially the same question as the reward. The results show broadly the same trends across methods as the main benchmark. These results can be found in Table 6 and Table 7 respectively.

Note that because remission time is conditional on remission having been achieved in a given episode, it does not make much sense to compare it between methods with significantly different remission rates.

### C.2    Computational Resources

This work was carried out using GPU workers on a workstation equipped with four Nvidia RTX6000 GPUs, each with 48 GiB of VRAM. We spent approximately 200 GPU hours on hyperparameter sweeps, 400 GPU hours training final models across all 8 environments, and 200 GPU hours on data restriction sweeps, totaling about 8½ days wall time. A large amount of compute was also spent on preliminary work.

Table 4: Online evaluation results for 8 structurally different *EpiCare* environments generated from environment seeds 1–8. For each variant, the standard deviation of the mean returns across 4 replicates is reported within the parenthesis.

| | BASELINES | | | TRAINED MODELS | | | | | | |
| | RAND | SoC | OP | EDAC | AWAC | BC | TD3+BC | IQL | DQN | CQL |
|---|---|---|---|---|---|---|---|---|---|---|
| MEAN | -0.1(0.7) | 39.3(0.7) | 95.2(0.0) | 2.4(15.9) | 17.1(1.7) | 18.5(1.7) | 52.3(10.8) | 72.5(1.0) | 75.2(0.8) | **78.0(0.9)** |
| ENV 1 | 8.3(1.5) | 47.8(1.0) | 95.7(0.0) | 7.2(17.5) | 30.1(1.8) | 24.4(1.0) | 71.3(5.2) | 76.5(0.7) | 77.0(0.6) | **79.4(0.7)** |
| ENV 2 | -2.8(0.7) | 37.4(0.4) | 94.0(0.1) | 0.9(4.1) | 19.4(2.2) | 22.1(2.5) | 68.9(1.8) | 73.8(1.7) | 75.6(0.9) | **77.8(0.3)** |
| ENV 3 | -13.1(0.1) | 34.5(0.8) | 94.7(0.1) | -6.2(31.5) | 8.0(1.1) | 11.0(1.5) | 10.6(22.2) | 68.5(1.8) | 72.4(1.3) | **75.6(0.8)** |
| ENV 4 | 1.0(0.7) | 35.9(0.4) | 95.6(0.0) | 2.6(12.0) | 20.0(1.0) | 20.4(1.9) | 69.2(1.5) | 71.0(0.8) | 75.8(1.4) | **78.8(1.2)** |
| ENV 5 | 7.9(0.7) | 36.0(0.4) | 95.4(0.1) | -4.9(14.1) | 21.4(2.1) | 22.0(1.2) | 36.5(24.8) | 72.1(0.7) | 74.1(0.6) | **78.2(1.0)** |
| ENV 6 | 0.5(1.0) | 45.6(0.8) | 95.9(0.1) | 6.7(11.4) | 22.2(3.2) | 18.5(3.1) | 57.6(8.8) | 72.9(0.7) | 78.2(0.5) | **80.0(0.5)** |
| ENV 7 | -1.1(0.4) | 42.4(1.3) | 94.9(0.0) | 0.1(20.8) | 9.9(1.6) | 14.3(1.2) | 55.1(6.0) | 72.8(0.7) | 74.5(0.5) | **77.9(0.8)** |
| ENV 8 | -1.2(0.6) | 35.0(0.6) | 95.3(0.0) | 12.9(16.0) | 5.9(0.3) | 15.5(1.4) | 54.1(16.0) | 72.2(0.9) | 73.8(1.0) | **76.6(1.6)** |

Table 5: Adverse event rates for the same baseline policies and models as in Table 4, in units of mean (standard deviation) of the number of adverse events per 10,000 trials.

|  | BASELINES | | | TRAINED MODELS | | | | | | |
|  | RAND | SOC | OP | EDAC | BC | AWAC | TD3+BC | DQN | IQL | CQL |
|---|---|---|---|---|---|---|---|---|---|---|
| MEAN | 55(9) | 31(5) | 2(1) | 46(13) | 45(12) | 36(8) | 31(10) | 25(8) | 22(11) | **20(10)** |
| ENV 1 | 31(2) | 19(2) | 0(0) | 42(25) | 42(17) | 21(7) | 17(4) | **12(3)** | 14(9) | 19(14) |
| ENV 2 | 71(9) | 41(6) | 11(6) | 61(14) | 55(19) | 63(8) | 36(14) | 34(17) | **21(12)** | 24(5) |
| ENV 3 | 54(5) | 28(4) | 0(0) | 42(8) | 49(14) | 23(4) | 41(14) | 25(8) | 17(9) | **13(6)** |
| ENV 4 | 36(7) | 15(1) | 3(2) | 38(18) | 24(7) | 28(9) | 19(6) | 21(7) | 22(10) | **13(12)** |
| ENV 5 | 58(7) | 24(6) | 1(1) | 47(9) | 47(12) | 21(4) | 25(9) | 30(9) | 24(7) | **18(8)** |
| ENV 6 | 74(16) | 42(14) | 3(2) | 46(15) | 56(10) | 32(15) | 44(15) | **18(5)** | 26(18) | 26(6) |
| ENV 7 | 65(23) | 36(4) | 0(0) | 51(12) | 50(4) | 62(8) | 36(17) | 37(11) | **27(16)** | 31(18) |
| ENV 8 | 47(3) | 42(6) | 0(0) | 43(4) | 39(13) | 36(12) | 24(2) | 24(2) | 21(6) | **18(9)** |

Table 6: Remission rate across 1000 episodes for each of the baseline policies and trained models.

|  | BASELINES | | | TRAINED MODELS | | | | | | |
|  | RAND | SOC | OP | EDAC | AWAC | BC | TD3+BC | IQL | DQN | CQL |
|---|---|---|---|---|---|---|---|---|---|---|
| MEAN | 0.47(0.01) | 0.72(0.00) | 1.00(0.00) | 0.47(0.12) | 0.55(0.01) | 0.59(0.01) | 0.80(0.07) | 0.91(0.01) | 0.94(0.00) | **0.95(0.00)** |
| ENV 1 | 0.52(0.01) | 0.76(0.01) | 1.00(0.00) | 0.49(0.14) | 0.64(0.01) | 0.62(0.01) | 0.91(0.03) | 0.93(0.00) | 0.95(0.00) | **0.96(0.00)** |
| ENV 2 | 0.47(0.00) | 0.72(0.00) | 1.00(0.00) | 0.51(0.02) | 0.58(0.01) | 0.63(0.02) | 0.91(0.01) | 0.92(0.01) | 0.95(0.00) | **0.96(0.00)** |
| ENV 3 | 0.41(0.00) | 0.67(0.01) | 1.00(0.00) | 0.46(0.21) | 0.45(0.01) | 0.55(0.01) | 0.56(0.12) | 0.89(0.01) | 0.93(0.00) | **0.94(0.00)** |
| ENV 4 | 0.47(0.01) | 0.67(0.00) | 1.00(0.00) | 0.46(0.15) | 0.56(0.01) | 0.59(0.01) | 0.90(0.01) | 0.89(0.01) | 0.94(0.01) | **0.95(0.01)** |
| ENV 5 | 0.50(0.01) | 0.70(0.00) | 1.00(0.00) | 0.39(0.10) | 0.59(0.01) | 0.60(0.01) | 0.69(0.17) | 0.91(0.00) | 0.94(0.00) | **0.96(0.01)** |
| ENV 6 | 0.46(0.01) | 0.75(0.00) | 1.00(0.00) | 0.43(0.08) | 0.55(0.02) | 0.57(0.02) | 0.81(0.06) | 0.90(0.00) | 0.95(0.00) | **0.96(0.00)** |
| ENV 7 | 0.44(0.00) | 0.76(0.01) | 1.00(0.00) | 0.44(0.15) | 0.55(0.01) | 0.56(0.01) | 0.82(0.04) | 0.92(0.00) | 0.95(0.00) | **0.96(0.00)** |
| ENV 8 | 0.47(0.00) | 0.69(0.00) | 1.00(0.00) | 0.54(0.13) | 0.48(0.00) | 0.57(0.01) | 0.82(0.10) | 0.91(0.01) | 0.94(0.00) | **0.95(0.01)** |

## C.3 Hyperparameter Sweeps

We ran hyperparameter sweeps for all RL methods for which we report performance. In all cases the hyperparameter sweep ranges were chosen in accordance with the ranges reported in their original papers, as described in Table 8. Additionally, EDAC and TD3+BC have an extra temperature hyperparameter not included in their original formulations due to the Gumbel-Softmax reparameterization.

Note that hyperparameters were swept on a grid of a few discrete values in order to save on computation. Although these values are representative of values reported in the literature, and other training runs outside the main sweeps did not reveal any regions of substantially greater performance, we expect that the small scale of hyperparameter sweeps limits the performance of the models somewhat. We view this as an acceptable tradeoff since we are presenting a novel benchmarking environment, not attempting to establish a hard limit on the possible performance of any of these methods.

The table omits two hyperparameters which were included on all models: FRAME STACK and PREVIOUS ACTION. These control the availability of previous observations and the last selected action respectively. We found that turning either of these features off negatively impacted performance, as expected, and did not explicitly include them in the sweep.

Table 7: Mean time to remission in episodes where remission was achieved for each of the policies (standard deviation across 4 replicates).

|  | BASELINES | | | TRAINED MODELS | | | | | | |
|  | RAND | SOC | OP | BC | EDAC | TD3+BC | AWAC | DQN | CQL | IQL |
|---|---|---|---|---|---|---|---|---|---|---|
| MEAN | 4.0(0.0) | 3.4(0.0) | 1.1(0.0) | 3.7(0.0) | 3.3(0.7) | 2.9(0.2) | 2.9(0.1) | 2.7(0.1) | **2.4(0.0)** | **2.4(0.0)** |
| ENV 1 | 4.0(0.0) | 3.4(0.0) | 1.1(0.0) | 3.7(0.0) | 3.3(0.4) | 2.6(0.1) | 3.1(0.1) | 2.7(0.1) | **2.3(0.0)** | **2.3(0.0)** |
| ENV 2 | 3.9(0.0) | 3.6(0.0) | 1.1(0.0) | 3.5(0.0) | 3.8(0.4) | 2.5(0.1) | **2.3(0.1)** | 2.5(0.1) | **2.3(0.0)** | 2.3(0.1) |
| ENV 3 | 4.1(0.0) | 2.9(0.0) | 1.1(0.0) | 3.6(0.1) | 3.2(0.9) | 3.7(0.6) | 2.5(0.1) | 2.6(0.1) | 2.5(0.1) | **2.4(0.0)** |
| ENV 4 | 4.0(0.0) | 3.2(0.0) | 1.1(0.0) | 3.6(0.1) | 3.6(1.3) | 3.0(0.2) | 3.1(0.0) | 2.7(0.0) | 2.5(0.1) | **2.4(0.1)** |
| ENV 5 | 4.1(0.0) | 3.4(0.1) | 1.1(0.0) | 3.7(0.0) | 3.5(0.5) | 2.7(0.1) | 2.9(0.1) | 2.8(0.1) | 2.5(0.0) | **2.4(0.0)** |
| ENV 6 | 4.1(0.0) | 3.8(0.1) | 1.1(0.0) | 3.7(0.0) | 2.7(0.9) | 3.0(0.1) | 3.3(0.1) | 2.7(0.0) | **2.5(0.0)** | **2.5(0.0)** |
| ENV 7 | 4.0(0.0) | 3.4(0.0) | 1.1(0.0) | 3.9(0.0) | 3.2(0.5) | 3.1(0.2) | 3.3(0.1) | 2.7(0.1) | **2.4(0.0)** | **2.4(0.0)** |
| ENV 8 | 4.1(0.1) | 3.4(0.1) | 1.1(0.0) | 3.7(0.1) | 3.3(0.9) | 2.9(0.2) | 3.1(0.1) | 2.7(0.1) | **2.5(0.0)** | **2.5(0.0)** |

Table 8: Hyperparameters used in the sweep for all benchmarked RL methods.

| ALGORITHM | HYPERPARAMETER | VALUES | OPTIMAL |
|---|---|---|---|
| AWAC | LAMBDA | 0.3, 1.0 | 0.3 |
| | LEARNING RATE | 1E-5 3E-4 | 3E-4 |
| | TRAINING STEPS | - | 2E5 |
| BC | LEARNING RATE | 3E-5, 1E-4, 3E-4 | 1E-4 |
| | TRAINING STEPS | - | 4E5 |
| CQL | ALPHA | 0.1, 0.25, 0.5, 1.0 | 1.0 |
| | GAMMA | 0.0, 0.1, 0.5, 0.9 | 0.0 |
| | $Q$ FUNCTION LEARNING RATE | 3E-5, 1E-4 | 3E-5 |
| | TRAINING STEPS | - | 2E5 |
| DQN | GAMMA | 0.1, 0.5, 0.9 | 0.1 |
| | $Q$ FUNCTION LEARNING RATE | 3E-5, 1E-4 | 1E-4 |
| | TRAINING STEPS | - | 2E5 |
| EDAC | ETA | 0.1, 1.0, 5.0 | 0.1 |
| | NUM CRITICS | 10, 55, 100 | 100 |
| | TEMPERATURE | 0.25, 1.0, 4.0 | 4.0 |
| | TRAINING STEPS | - | 5E5 |
| IQL | TAU | 0.5, 0.7, 0.9 | 0.9 |
| | BETA | 3, 6, 10 | 3 |
| | ACTOR DROPOUT | 0.0, 0.1 | 0.1 |
| | TRAINING STEPS | - | 5E5 |
| TD3+BC | ALPHA | 1.0, 2.5, 4.0 | 4.0 |
| | TEMPERATURE | 0.3, 1.0, 3.0 | 3.0 |
| | TRAINING STEPS | - | 5E5 |

For the number of training iterations, no list of values is given, because it was not part of the grid search. Instead of explicitly varying the value of this parameter, we performed long training runs, logging evaluations throughout. For final training, we used the number of training steps where the training curves exhibited maximum performance. In some cases, such as for CQL, this value needed to be drastically reduced below previously reported values.

### C.4   Data Availability Dependent Hyperparameters

We have observed that for some models, the optimal hyperparameters are dependent on the size of the training dataset. Figure 11 gives an example of this for IQL, where the optimal number of training iterations to perform depends in a clear way on data availability. Each of the curves on the left shows the evaluation performance (smoothed with an exponential moving average, $\alpha = 0.01$) of the trained model. In each case, the model begins to overfit after an initial peak in its evaluation performance, so the number of training steps depends on the training set being considered. Interestingly, the location of this peak appears to be almost exactly proportional to the dataset size.

### C.5   Data Restriction Sweep Continued

In addition to measuring the mean returns as a function of training data availability (Figure 4), we also investigated the effect of limited training data on the safety of each method, quantified by the adverse event rate. We found that in general, although training data availability was very important to performance, it had little effect on the ability of RL methods to avoid adverse events. This is unsurprising given that these models are optimizing mean rewards and do not have any features specifically intended to avoid adverse events.

### C.6   Patient-Specific Effects

As an additional complication to the model, we ran a separate experiment considering the presence of patient-specific effects which varied the details of the *EpiCare* on a patient-to-patient level.

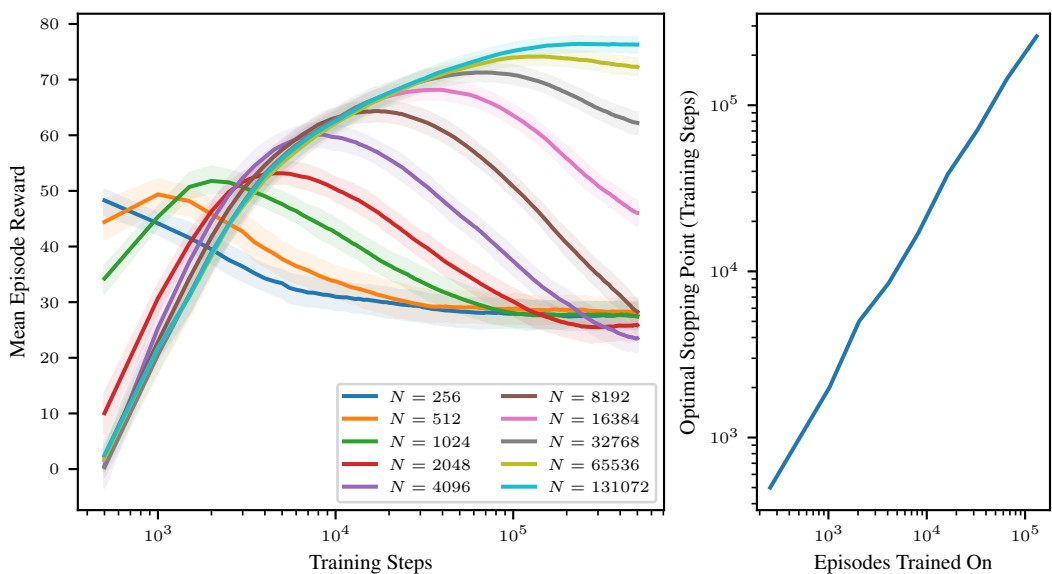

Figure 11: Left: normalized performance over the number of training steps for IQL. Shaded area corresponds to the minimum and maximum. Mean, minimum, and maximum were exponentially smoothed with $\alpha = 0.01$. Right: The optimal stopping point (peak of the curves on the left) as a function of the size of the training set.

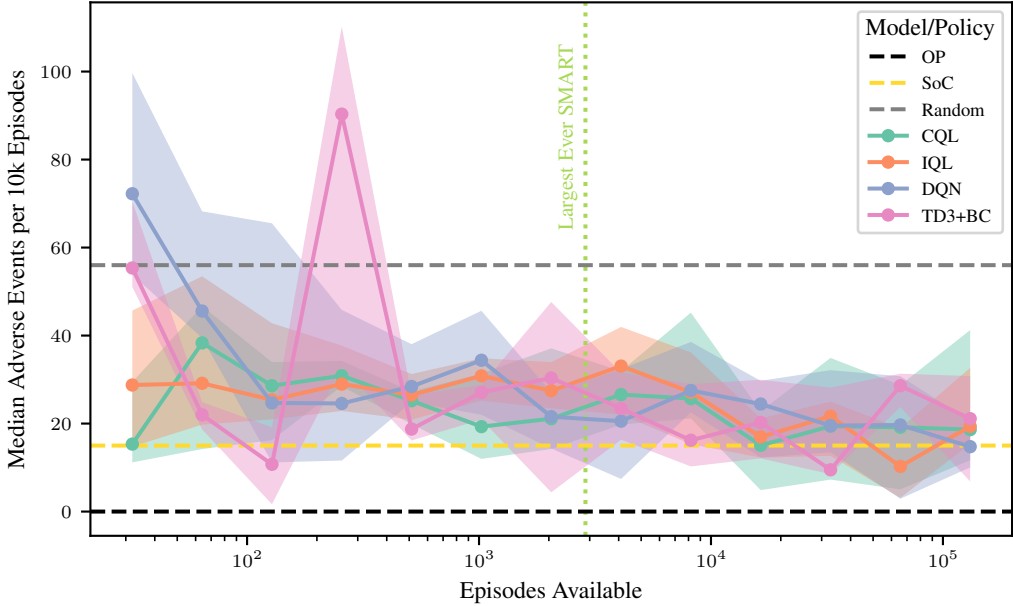

Figure 12: Data restriction trials for the adverse event rate of the four top performing RL models, compared to three baselines policies, whose median per-episode performance is dictated only by the environment parameters and not by data availability. TD3+BC again shows double-descent-like behavior at 256 Episodes Available.

Table 9: Online evaluation results in terms of mean returns on models trained using a restricted training set consisting of only 2,048 episodes episodes from environment 1. The standard deviation of the mean returns across four replicates is reported within parentheses.

|  | CQL | IQL | DQN | TD3+BC |
|---|---|---|---|---|
| ENV 1 | 38.1(3.5) | 25.5(1.1) | 52.9(3.0) | 15.4(50.9) |

Table 10: Performance of CQL decreases significantly when patient-specific effects are included. Mean returns are given together with the standard deviation across four replicates, as in the main text.

|        | Original     | w/ Patient-Specific Effects |
|--------|--------------|------------------------------|
| Env 1  | 78.0 (0.9)   | 65.20 (0.74)                 |
| Env 2  | 77.8 (0.3)   | 62.13 (0.79)                 |
| Env 3  | 75.6 (0.8)   | 55.89 (0.91)                 |
| Env 4  | 78.8 (1.2)   | 59.37 (0.75)                 |
| Env 5  | 78.2 (1.0)   | 60.17 (0.77)                 |
| Env 6  | 80.0 (0.5)   | 61.33 (0.77)                 |
| Env 7  | 77.9 (0.8)   | 61.68 (0.76)                 |
| Env 8  | 76.6 (1.6)   | 58.23 (0.77)                 |
| Mean   | 77.36 (1.07) | 60.50 (0.78)                 |

We modeled patient-specific effects in four separate ways. First, we introduced a patient-specific transition modification vector $\mathbf{m}_p$ such that

$$T(\mathrm{s}_j|\mathrm{s}_i, a, p) = (1 - T(\mathrm{s}_\mathrm{r}|\mathrm{s}_i, a) - T(\mathrm{s}_\mathrm{a}|\mathrm{s}_i, a)) \frac{(\mathbf{m}_p)_j(\mathbf{m}_a)_j \mathbf{T}_{i,j}}{\sum_{k=1}^{n_s}(\mathbf{m}_p)_k(\mathbf{m}_a)_k \mathbf{T}_{i,k}}. \tag{6}$$

When patient-specific effects were included, the entries of the vector $\mathbf{m}_p$ were sampled from the uniform distribution over the interval $(0.25, 1.75)$.

Second, we included patient-specific remission modifiers to titrate how likely any given patient was to achieve remission. This was modeled as an action-indexed multiplier to up-regulate or downregulate the probabilty any given action would lead to remission on a patient-specific basis. For this experiment, these modifiers were also drawn from the interval $(0.25, 1.75)$.

Third, we included a patient-specific adverse-event modifier which set the adverse-event threshold $o_\mathrm{a}^*$ on a patient-to-patient basis rather than selecting a single value for all episodes. This modifier was drawn from the range $(0.999, 1/o_\mathrm{a}^*)$ so that each patient's adverse event threshold would range from 0.998001 to exactly 1.

Finally, we included patient-specific symptom modifiers which add patient-specific fluctuations to the action-based observation confounding vector $\delta_a$. These are drawn from the same range as the treatment-specific symptom modulation.

As a result of these various patient-specific effects, the environment becomes substantially more difficult and may be even more representative of the challenge of real longitudinal care scenarios, where patients are known not to be homogeneous even when their disease state is identical. Unsurprisingly, this increased difficulty poses a greater challenge for RL approaches to the problem, as shown for CQL in Table 10. Mean returns decrease by an average of over 16 points, on the same order of magnitude as several episodes of failed treatment. This suggests that even when RL algorithms appear to perform well on the original *EpiCare* benchmark, proper handling of patient-specific effects will be an important milestone before considering translational applications of those algorithms.

## D   Off-policy Evaluation Continued

Full scatter plots of the data from which the RMSE of Table 1 was calculated are given in Figure 13. Each data point represents the eight-fold bootstrapped mean of one OPE estimator for the performance of a single fully trained model, of which there are a total of 32: 4 replicates across 8 distinct environments.

In addition to the RMSE of OPE being quite high as reported above, it is also important to note that there is no visible relationship between the OPE estimate and the true online returns. In a few cases, a small cluster of scores was predicted for all models in the category, but only for AWAC and BC was this cluster neard the $x = y$ line indicating an unbiased estimate. This is quantified using Pearson correlation coefficients in Table 11. Broadly the same performance trends appear in these results as in the RMSE.

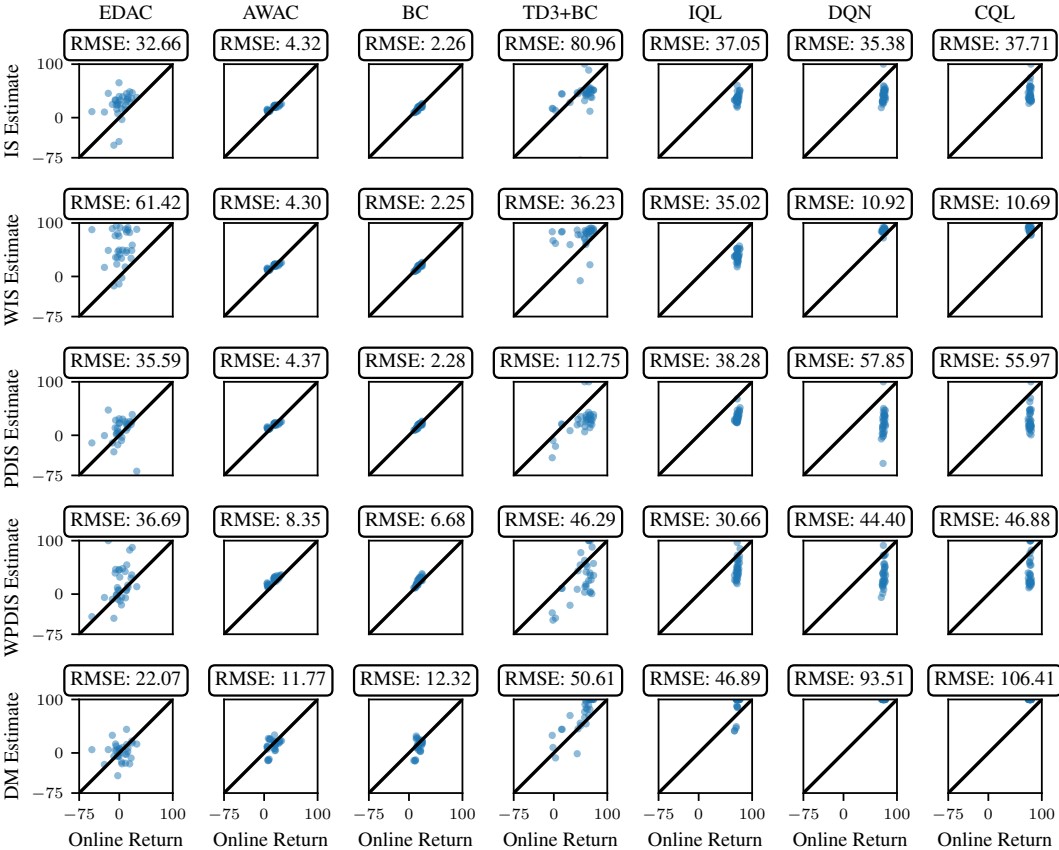

Figure 13: OPE estimates plotted against true mean normalized rewards for every combination of OPE method and RL model with RMSE reported. OPE estimates greater than 100 are truncated in this figure for display purposes.

Because we bootstrapped the importance sampling estimators, we can also investigate the high variance known to affect them [30]. The distribution of the bootstrap predicted variance for each estimator is plotted in Figure 15. Many of these values are on the same order of magnitude as the reward itself, even despite the fact that the test set contains over $10^5$ episodes. Note also that although bootstrapping reduces variance, it can introduce bias, which we expect to be the reason for the apparently systematic errors visible in Figure 13 despite the importance sampling methods being unbiased estimators.

We suspected that the reason that OPE methods perform better on AWAC and BC than on other models might be due to AWAC selecting actions more similar to the training data. To investigate this, we computed the geometric mean of the probability that each model would perform the same action as was selected by the SMART policy in the training data. Indeed, we find that this action probability is a good predictor of the RMS error of OPE. The rank order of average training action probability is nearly identical to the rank order of mean RMS error across OPE value estimates (Figure 14).

Another potential problem with OPE in the context of *EpiCare* is the large positive and negative rewards of the two terminal states; importance sampling methods have trouble with sparse rewards in general [13] and do not explicitly model episode termination [29]. Indeed, the reward predictions of all five OPE methods very frequently exceeded the maximum possible episode reward of 100, often by a substantial margin. This can lead to extremely large RMSE values. In particular, the evaluation of *every* DQN and CQL model by DM was greater than 100. This may be the source of the apparent bias in the theoretically unbiased importance sampling estimators (IS, WIS, PDIS, and WPDIS).

When observations are non-Markovian (as in this case due to partial observability), the error of importance sampling-based OPE methods is known to scale exponentially with the horizon [30]. This is likely a smaller effect in our environment due to the episode lengths being quite short.

Table 11: Pearson correlation between the OPE estimates and the true online returns evaluated on 1,000 episodes for each combination of OPE method and RL model, across 8 seeds with 4 replicates, as in Table 1.

|        | EDAC | AWAC | BC   | TD3+BC | IQL  | DQN  | CQL   |
|--------|------|------|------|--------|------|------|-------|
| IS     | 0.31 | 0.91 | 0.90 | 0.15   | 0.41 | 0.31 | 0.17  |
| WIS    | 0.11 | 0.91 | 0.90 | 0.12   | 0.25 | 0.20 | −0.33 |
| PDIS   | 0.00 | 0.90 | 0.89 | 0.17   | 0.51 | 0.31 | 0.13  |
| WPDIS  | 0.34 | 0.90 | 0.88 | 0.55   | 0.53 | 0.44 | 0.19  |
| DM     | 0.12 | 0.43 | 0.47 | 0.71   | 0.44 | 0.18 | 0.15  |

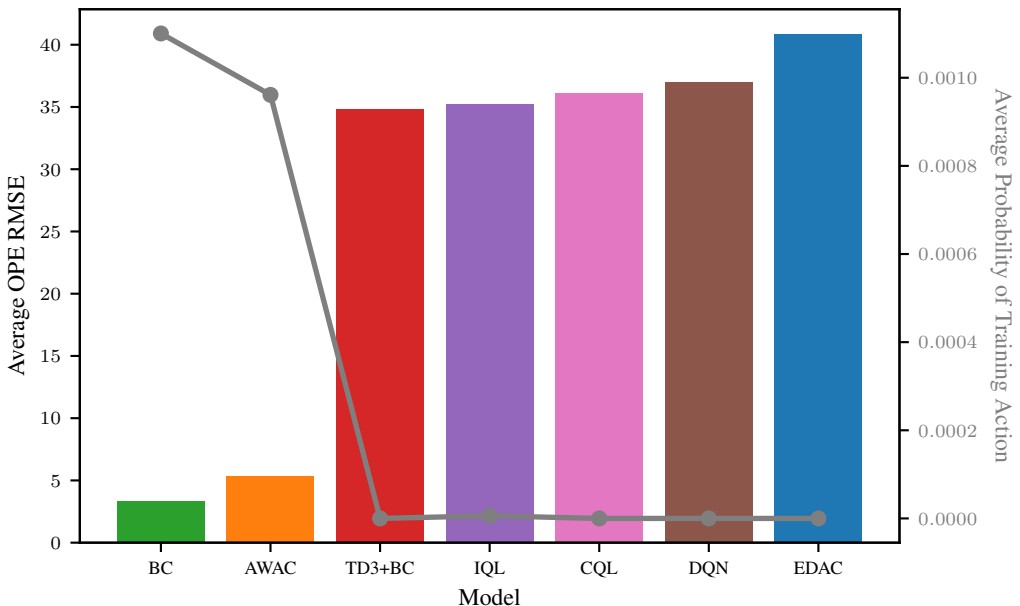

Figure 14: Average RMS error between OPE estimates and online evaluation results (bars) compared to the geometric mean of the probability that each policy chooses the same action that was taken in its training data (gray line).

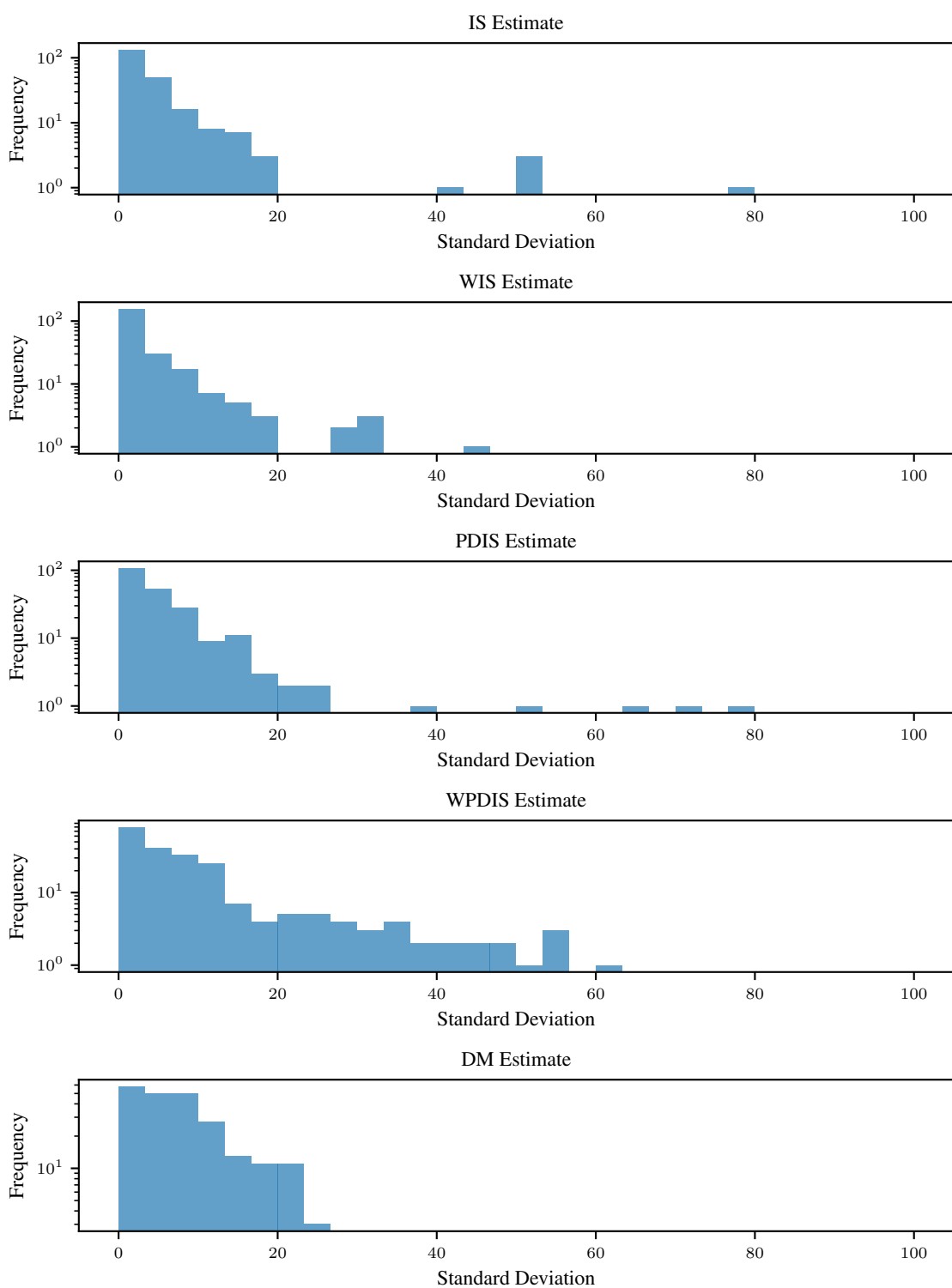

Figure 15: The frequency of the standard deviation of bootstrapped estimates for each of the OPE methods. Additional outlier standard deviation values exist which are not plotted, those being 143.7 and 458.3 for IS and 116.1, 692.2, and 110.2 for PDIS.

