# OpenReview forum: "EpiCare: A Reinforcement Learning Benchmark for Dynamic Treatment Regimes"
_NeurIPS.cc/2024/Datasets_and_Benchmarks_Track — NeurIPS 2024 Track Datasets and Benchmarks Poster_

### Official Review · Reviewer_h9aa · 2024-07-25
**The first work to bridge the gap between medical RL and real-world application.**

**Rating:** 7
**Confidence:** 3
**Correctness:** Yes
**Clarity:** Yes

**Review:**

The paper introduces the EpiCare benchmark, simulating real-world challenges in healthcare RL applications and provides thorough testing of the latest offline RL models and OPE techniques, ensuring robust methodologies. It clearly describes the inherent challenges in healthcare RL, such as sparse rewards and heterogeneous treatment effects, and effectively communicates the limitations of offline RL in data-limited scenarios. The EpiCare benchmark fills a critical gap by offering a standard for evaluating RL in longitudinal healthcare scenarios, advancing the field by exposing the shortcomings of current OPE methods.

This work is crucial as it elucidates the challenges of applying RL in healthcare, setting the stage for future research and advancing practical applications in medical RL.

Pros:

	•	Introduces EpiCare to simulate healthcare challenges.
	•	Rigorous testing of advanced RL and OPE techniques.
	•	Clearly presents results and limitations in healthcare RL.
	•	Advances medical RL by highlighting evaluation gaps.

Cons:

	•	Focuses narrowly on specific healthcare applications.
	•	May face challenges in generalizing across diverse settings.
	•	Depends on simulations that may not capture real-world complexities.

**Strengths:**

See review section

**Additional Feedback:**

N/A

**Documentation:**

Yes

**Ethics:**

No ethical concerns.

**Opportunities For Improvement:**

N/A

**Relation To Prior Work:**

Yes

**Summary And Contributions:**

The submission introduces the EpiCare benchmark, designed to replicate challenges in applying reinforcement learning (RL) to longitudinal healthcare settings. This benchmark is the first work aims to mimic scenarios with low data availability, short treatment episodes, sparse rewards, partial observations, and heterogeneous treatment effects. This paper also evaluate the performance of well-known RL techniques, demonstrating that several OPE techniques commonly used in medical RL literature do not perform adequately under the simulated conditions

---

> ### Author Rebuttal · Authors · 2024-08-17
>
> Thank you for your thoughtful feedback. We appreciate the opportunity to clarify our work.
>
> While our focus may appear narrow, Dynamic Treatment Regimes (DTRs) actually represent a broad, crucial area in healthcare that currently lacks a standardized benchmark. Our focused approach allows us to provide meaningful insights into the challenges of applying RL to healthcare. EpiCare captures essential complexities, enables relevant evaluation, fills a critical gap, and balances generality with specificity. A more general benchmark might become too broad to adequately represent any specific medical application.
>
> Regarding generalization, EpiCare uses a flexible POMDP framework with configurable parameters. The core challenges it captures (partial observability, delayed rewards, heterogeneous treatment effects) are common across many health scenarios. The environment is scalable in complexity, allowing testing under various conditions. Our OPE results have broad implications beyond DTRs. While EpiCare may not capture every nuance of all possible healthcare scenarios, it serves a critical niche in providing a standardized benchmark for longitudinal care decisions.
>
> We understand the concern about capturing real-world complexities. EpiCare encapsulates key challenges in longitudinal healthcare: partial observability, heterogeneous treatment effects, sparse/delayed rewards, and complex state transitions. It provides a controlled setting for evaluating RL algorithms and OPE methods on these core challenges. Our goal is to test fundamental capabilities required for RL in healthcare, not to replicate specific diseases. We view EpiCare as a complement to, not a replacement for, studies on real-world data (as, e.g., Half Cheetah complements but does not replace real-world robotics RL applications).
> We will clarify these points in our revision to better communicate EpiCare's intended role and value in advancing RL for healthcare applications.

---

### Official Review · Reviewer_E79g · 2024-07-25
**Simulator longitudinal healthcare problems - but no grounding in real data and no clear parameter learning rule.**

**Rating:** 6
**Confidence:** 3
**Clarity:** The paper is well written.

**Review:**

The authors present a simulation framework, that is, an environment with several configurable parameters that can be used for RL in healthcare problems. However, this environment is very abstract and to use it for any practical use case would need a lot of effort in terms of data collection, environment parameter estimation and then testing with ground truth data. The authors have also not factored patient-specific characteristics in their environment. The state, action and observation spaces considered in this work seem to be fairly limited to be of use in real-world problems.

**Strengths:**

The authors present an interesting problem, one that is of utmost importance to help RL become useful in real-world healthcare applications.

**Additional Feedback:**

1. Why do the authors call their simulated environment a "benchmark"?
2. What are the definitions of "longitudinal healthcare applications" and "longitudinal patient care"?
3. When the authors are developing a simulation environment, why do they generate off-policy datasets? Why not use online RL with this simulator?
4. When the authors state that "To this end we have chosen some specific environment hyperparameters in close collaboration with medical professionals which reflect the realities of longitudinal patient treatment scenarios." in Line 72, how have they validated this? For new diseases will this exercise have to be conducted again using medical experts?
5. The state space assumed by the authors only involves the current disease state. Aren't there other body parameters that are important to determine future health state of the patient?
6. Why does the state space not include any patient-specific properties including medical history?
7. Why is the Oracle Policy considered to be greedy/myopic?
8. Why is the SoC policy also a greedy/myopic one?
9. Why does the SoC policy consider each individual episode as a "non-stationary multi-armed bandit"? Aren't there multiple treatment steps in an episode?
10. Does the SMART policy consider multi-step treatments?
11. The observations by the authors that the OPE methods do not perform well are with respect to their simulated environments. How does this relate to any real-world dataset?
12. Why don't the authors learn the parameters of their simulator from real-world medical datasets?

**Correctness:**

Since the paper only proposes a mathematical framework for simulation, it is tough to comment on its correctness. The authors have not compared the predictions of their simulator with any real-world data.

**Documentation:**

The authors have given code for their simulator and associated documentation. But since they do not release any new datasets, there is no documentation on that.

**Ethics:**

The ethical concerns in this paper are standard as those regarding use of simulators for healthcare. However, the authors do not provide any specific dataset, benchmark or recommendation, hence the impact is fairly limited.

**Limitations:**

The authors have discussed the limitations of their work in a separate section.

**Opportunities For Improvement:**

The simulator environment presented by the author is too generic. It is not immediately clear as to how this will be useful for real-world use cases. The simulator does not seem to be grounded with real patient data. Also, when defining the healthcare problem over a longer time frame, it is not clear if the state considered by the authors covers all relevant health related aspects instead of only covering a specific disease.

**Relation To Prior Work:**

The authors cover several works in literature that study RL in healthcare and also some that use simulators. They claim that their work is novel as none of the previous works have simulators designed for "longitudinal healthcare applications".

**Summary And Contributions:**

The authors present a simulation environment for longitudinal healthcare applications. This environment follows the OpenAI Gym format and therefore is compatible with most RL libraries. Furthermore, the authors also make the environment customizable with adjustable parameters. They also show using their simulator that current off-policy evaluation methods used in RL don't work well for longitudinal healthcare use cases.

---

> ### Author Rebuttal · Authors · 2024-08-17
>
> W: weakness, Q: question.
>
> ## Q1
> EpiCare is not intended as a disease simulation but rather as a configurable benchmark to measure the readiness of RL and OPE methods for healthcare challenges. Similar to Half Cheetah in continuous control, EpiCare lacks an exact real-world analogue but captures essential challenges. This allows rigorous testing of RL and OPE methods, crucial for safe deployment in healthcare but impossible to validate comprehensively with real-world data due to missing counterfactuals [1].
>
> [1] Gottesman, Omer, et al. "Evaluating reinforcement learning algorithms in observational health settings."
> unable to model hidden states.
>
> ## Q2
> Longitudinal healthcare refers to medical scenarios where treatment decisions are made sequentially over multiple time points, allowing for the patient's condition to evolve and requiring adaptive treatment strategies. This approach contrasts with acute care, where the focus is on non-sequential short-term interventions.
>
> ## Q3
> See W2 in the rebuttal for reviewer CJAm
>
> ## Q4
> The environment hyperparameters were chosen by hosting “learning” sessions with doctors and asking questions about the expected ranges for these general parameters for a moderately complex disease. This is not a process that needs to be repeated, as the goal is not to tune this environment to model any specific disease.
>
> ## Q5&6
> As you have seen, our initial attempt at a simplified but representative benchmark only models differences between patients in the form of “communicating classes” of hidden states, which could correspond to, for example, a model of chronic depression where male and female patients experience similar states but the symptoms and transition probabilities are not identical for the two subgroups.
>
> As a response to this review, we have added optional patient variation in key transition and observation probabilities. As might be expected, this makes the problem more difficult, underscoring our main point about the need for further development of algorithms before considering real longitudinal medical applications. See the general rebuttal for a table containing these results.
>
> ## Q7
> The Oracle Policy is myopic because it chooses the action which is most likely to yield the highest reward at each time step (without looking multiple time points ahead).
>
> ## Q8
> Since SoC is intended to represent a “state-agnostic” clinician unable to model hidden states, the SoC cannot base any of its decisions on future hidden state dynamics and thus must act greedily.
>
> ## Q9
> Yes, there are multiple treatment steps in each episode. There is no conflict between there being multiple treatment steps in an episode and each individual episode being a Multi Armed Bandit (MAB).
>
> ## Q10
> We consider each step within an episode to not necessarily have a fixed length in terms of time. Instead, a step represents a single treatment decision, that could include the decision to begin a course of treatments necessitating multiple visits. Of course there are any number of complications such as terminating a treatment course early which we do not attempt to model. However, EpiCare is the first benchmark based on longitudinal medical care, and is already difficult even without these extra complications.
>
> ## Q11
> EpiCare serves as a benchmark to validate the performance of Off-Policy Evaluation (OPE) by enabling comparisons between online and offline evaluations, something that real datasets cannot achieve due to ethical concerns and missing counterfactual data. By creating a simplified benchmark akin to longitudinal healthcare datasets, EpiCare provides a "minimum bar" for performance, indicating that if RL or OPE methods fail on EpiCare, they are unlikely to succeed in more complex real scenarios—similar to how failing on the Half Cheetah benchmark suggests poor performance in corresponding real-world robotics problems.
>
> ## Q12
> As explained in more detail above, EpiCare is a benchmark which is intended to represent the challenges of longitudinal care scenarios rather than model any one specific disease. It's unethical to represent it as a platform for sim2real transfer in light of all the necessary complexities of real medical care which we do not model. Therefore, we purposefully avoid attempting to fit the parameters to any specific dataset.
>
> ## W1
> Evaluating RL and OPE performance with ground truth data is impossible in real-world scenarios due to missing counterfactuals. There is currently a “Catch 22” in the literature. RL cannot be applied to real patients because it has never been validated on real patients, and no one will attempt to validate it on real patients until RL is satisfactorily performant on medical-style datasets. EpiCare begins to bridge this gap, providing the first medical-style dataset for which RL algorithms can be evaluated online.
>
> ## W2
> See Q5&6
>
> ## W3
> While we agree that grounding the simulator with real patient data could provide additional realism, doing so would limit its generalizability and potentially introduce biases specific to the dataset used. Our goal is to provide a "minimum bar" (like Half Cheetah) for algorithm performance: any algorithm that cannot perform well on EpiCare is certainly not ready for real longitudinal care applications.
>
> ## W4
> EpiCare models a single patient population whose states represent various stages or manifestations of a disease, including comorbidities and complications.
>
> ## W5
> We appreciate the reviewer’s concern about the lack of comparison with real-world data. However, it's crucial to understand that the primary purpose of EpiCare is not to accurately simulate any specific disease, but to provide a benchmark for RL and OPE methods in longitudinal healthcare settings. Our framework allows us to answer important questions about OPE efficacy and RL performance that cannot be definitively answered with real-world data alone. We view EpiCare as a complementary tool to real-world studies, not a replacement for them.

---

> > ### Comment · Reviewer_E79g · 2024-08-22
> >
> > I thank the authors for their detailed responses to my questions. Here are some additional comments:
> >
> > 1. "See W2 in the rebuttal for reviewer CJAm": I did not understand how this answers my question on use of online vs. offline RL here.
> > 2. "Yes, there are multiple treatment steps in each episode. There is no conflict between there being multiple treatment steps in an episode and each individual episode being a Multi Armed Bandit (MAB).": Then what do the "arms" represent in this case- actions or policies?
> >
> > Based on the responses by the authors and the other reviews, I have raised my score.

---

> > > ### Author Response · Authors · 2024-08-27
> > >
> > > Thank you very much for your additional questions, see our response below.
> > >
> > > *"See W2 in the rebuttal for reviewer CJAm": I did not understand how this answers my question on use of online vs. offline RL here.*
> > >
> > > Our apologies, we misinterpreted your question as being about using an Online RL agent as a behavior policy to collect the online dataset. In terms of online vs. offline RL algorithms, the reason for generating offline datasets is that we are chiefly interested in evaluating offline RL. Due to the inherent risks associated with online RL, namely needing to deploy a tabula-rasa algorithm on patients, any insights we derive on online RL performance with our simulation would be practically useless, as it would never justify the deployment of online RL in the field. At best, one could evaluate offline RL with online fine-tuning after the fact, which many of the algorithms we tested are capable of (CQL, AWAC, IQL). This would be interesting to do, but our main concern is determining the “baseline” initialized performance after being trained on an offline dataset collected in a manner consistent with clinical practice (i.e SMART).
> > >
> > > *"Yes, there are multiple treatment steps in each episode. There is no conflict between there being multiple treatment steps in an episode and each individual episode being a Multi Armed Bandit (MAB).": Then what do the "arms" represent in this case- actions or policies?*
> > >
> > > The arms of the MAB each represent the treatment options. At every time step an arm is pulled, that is, a treatment is selected. Then a reward is tendered, transitions occur, and observations are emitted, bringing the system to the next time step. We will rework the text of the paper to make this more clear.

---

> > > > ### Comment · Reviewer_E79g · 2024-08-31
> > > >
> > > > Thanks for these clarifications.

---

### Official Review · Reviewer_CJAm · 2024-07-31
**The paper presents a novel benchmark designed to evaluate offline reinforcement learning (RL) methods and off-policy evaluation (OPE) techniques in healthcare, specifically for dynamic treatment regimes.**

**Rating:** 6
**Confidence:** 3
**Correctness:** The technique of the work is sound.
**Clarity:** Yes.

**Review:**

Strengths:

1. The paper is well organized, and the writing is clear and well-understood.
The author clearly introduces the details of the environment design, including the state transition function and the design of the reward function, and details the policies used, including the oracle policy and the policy for collecting data.

2. The authors propose a novel benchmark for dynamic treatment regimes that provides additional evaluation tools and potential research directions for the future development of medical RL and offline RL.

3. The authors conduct extensive experiments using five offline DRL and five OPE methods.
The results show that these methods have not yet arrived at satisfactory results, thus providing further insights and evaluation tools to improve these methods.


Weaknesses:

1. The authors used the SMART policy to collect the data, but why not use a policy trained by online RL methods to collect the data, as done in D4RL[1]?

2. Are there any methods for measuring the quality of the data in the dataset? I think it would be helpful if the authors could provide the results of the SMART policy.

3. The experimental section lacks a discussion on the effect of offline RL with datasets of different qualities. For example, in D4RL, the mujoco environments contain a variety of datasets of different qualities, including expert, medium, replay, and random levels.

4. As a benchmark specifically for use in healthcare, it is important to know how to use its data and evaluation results in real-world scenarios.
It would have been helpful if the authors could provide more analysis of the data and results to show how the benchmark could help in real-world medical scenarios.
For example, how close the generated data is to the real data, whether the trained policy can be used in real treatments, and some sim2real experiments.

[1] Fu, Justin, et al. "D4rl: Datasets for deep data-driven reinforcement learning." arXiv preprint arXiv:2004.07219 (2020).

**Strengths:**

Please see the strengths above.

**Additional Feedback:**

Please see the weakness in the review part.

**Documentation:**

Yes.

**Ethics:**

Not applicable.

**Limitations:**

Not applicable.

**Opportunities For Improvement:**

It would have been helpful if the authors could provide more analysis of the data and results to show how the benchmark could help in real-world medical scenarios.

**Relation To Prior Work:**

The work discussed the difference from existing works.

**Summary And Contributions:**

The paper presents a novel benchmark designed to evaluate offline reinforcement learning (RL) methods and off-policy evaluation (OPE) techniques in healthcare, specifically for dynamic treatment regimes.
The authors designed the benchmark with four main considerations: 1) realistic difficulty, 2) patient safety, 3) reproducibility and configurability, and 4) standardized benchmarks.
The authors evaluated five offline RL methods and five OPE methods on this benchmark. They demonstrated that offline RL methods suffer from low data settings, and OPE techniques also fail to perform adequately under simulated conditions.

---

> ### Author Rebuttal · Authors · 2024-08-17
>
> In response to your valuable feedback, we are making several key improvements to our paper. We're clarifying our rationale for using the SMART policy for data collection, expanding our discussion on dataset quality effects, and providing more comprehensive performance comparisons. We're also adding clearer guidance on interpreting EpiCare results in real-world contexts, including performance at realistic data levels and comparisons with real-world datasets.
>
> Below W denotes weakness.
>
> ## W1
>
> Our decision to use SMART as our behavior policy was deliberate and rooted in the realities of medical data collection and the unique challenges of applying RL in healthcare settings.
>
> 1. **Realism in Medical Data Collection:** The SMART policy simulates a sequential multiple assignment randomized trial, which is a common method for longitudinal multi-treatment studies in medicine. This approach better reflects the types of data that would be accessible to offline RL algorithms in real-world medical applications, where patient privacy considerations limit alternate data collection methods.
>
>
> 2. **Differences from Robotics Domains:** While D4RL uses online RL policies for data collection in robotics applications, this approach is less applicable in medical settings. In robotics, training data often comes from programmed controllers producing near-optimal trajectories. However, in healthcare, we rarely have access to such "optimal" data, and the goal is often to improve upon current standards of care rather than to replicate existing near-optimal strategies.
>
>
> 3. **Effects on RL Performance:** Interestingly, training on data from better-performing policies can sometimes hinder performance, as such data may result from entrenched policies that don't sufficiently explore the state space. This limited exploration can prevent offline RL algorithms from learning novel, potentially more effective strategies. We demonstrate this effect in Table 8 of our appendix, where we compare policies trained on SMART data versus those trained on Standard of Care (SoC) data. As per the reviewer’s recommendation, we will also add models trained on the data generated by an online RL algorithm to this table in order to provide a more complete picture of how the behavior policy affects RL performance.
>
> We will update the paper to make these points more explicit, adding Table 8 to the main body of the text, clarifying our rationale for using the SMART policy, and adding more discussion of the effects data quality has on model performance.
>
> ## W2
>
> We measure the quality of the datasets in terms of the performance of the policies from which the datasets are collected. At present, the performance of SMART can be seen in Figure 7 and is discussed in Appendix B3. To bring the issue of dataset quality more to the foreground, we will include SMART as one of the baselines in the tables throughout the paper.
>
> ## W3
>
> A very important point. Indeed, the policy used to generate training data has a substantial effect on the performance of offline RL methods trained on that data. One important reason for this is that policies which perform better actually explore a smaller region of the state space because they achieve remission faster. We will include some new discussion on this topic in the main body of the paper. Also see Appendix Table 8, to which we will add data for the performance of policies trained with data collected from an online RL behavior policy. Outside of varying data quality by changing the behavior policy as previously discussed, we use data availability asanother “difficulty knob”. This knob is important because medical data tends to be quite limited. Note in particular that our main results table is for the “easy” version where a huge number of training episodes are available. At realistic levels of data availability, performance does not consistently beat the state-agnostic SoC. The revised MS will include a results table for offline RL agents trained using more realistic data quantities.
>
> ## W4
>
> We acknowledge that clearer guidance is needed on interpreting and using EpiCare results, and we will address this by adding a guidance section to the revised paper. To clarify our vision:
>
> 1. **Benchmark Purpose:** EpiCare serves as a "minimum bar" for RL algorithms in medical decision-making. It's designed to guide research towards addressing key challenges like sparse rewards, low data availability, partial observability, and heterogeneous treatment effects. It is not intended to be parametrized in order to actually implement sim2real for any particular disease.
>
> 2. **Conceptual Framework:** EpiCare can be thought of as something like a "Half Cheetah" for medicine - not a perfect representation of any specific disease, but a benchmark to drive algorithm development.
>
> 3. **Sim2Real Limitations:** Traditional sim2real experiments are not feasible in medical scenarios due to ethical and counterfactual constraints, limiting our ability to directly validate on real patients.
>
> 4. **Future Applications:** Our ultimate goal is to enable human-in-the-loop RL systems for clinical decision support. EpiCare aims to facilitate the development of algorithms that can be trained offline to provide useful treatment recommendations in such contexts. In this way we see our work as a necessary stepping stone towards this eventuality.
>
> 5. **Complexity and Realism:** While EpiCare is a simplified version of real-world medical problems, it captures essential characteristics and reveals important existing limitations in OPE. We will be adding comparisons with real-world datasets like MIMIC-IV to illustrate similarities in complexity.
>
> We believe that by "cracking" EpiCare, the RL community will develop algorithms capable of providing valuable clinical decision support. Benchmarks like ours are a crucial step along the road realizing the potential of RL in healthcare, and we will provide clearer guidance on interpreting its results in this context

---

### Author Rebuttal · Authors · 2024-08-17

## General Rebuttal

We thank all the reviewers for their thorough and insightful feedback on our paper introducing EpiCare. Your comments are invaluable in helping us improve the clarity and impact of our work’ and we appreciate the time and effort you've invested in reviewing our manuscript.

### Common Themes

1. **Purpose and Nature of EpiCare:** We recognize that there was some confusion about the nature of EpiCare. To clarify, EpiCare is not intended to be a disease-specific simulator, but rather a benchmark for evaluating reinforcement learning (RL) algorithms and off-policy evaluation (OPE) methods in longitudinal healthcare settings. We will add a new section to our manuscript to make this distinction clear and to provide better guidance on how to interpret and use EpiCare. Note in particular there are significant ethical concerns with testing RL algorithms on real patients, yet this is ultimately the only way to measure real world performance as it is impossible to observe the (counterfactual) results of the actions which RL algorithms might have taken. This is the motivation for OPE as a field, but OPE *itself* must be validated; this need, motivating our introduction of a benchmark environment. Our results show OPE methods currently fall short.

2. **Realism and Complexity:** While EpiCare is indeed a simplification of real-world medical scenarios, it captures essential challenges such as partial observability, heterogeneous treatment effects, and sparse rewards. We see it as analogous to a benchmark like the Half Cheetah problem in continuous control: not representative of any one real-world scenario, but presenting similar challenges, such that it forms a “minimum bar” for algorithm performance. RL algorithms that cannot beat EpiCare are very likely to perform poorly in real-world settings, as are OPE methods which cannot accurately predict RL performance on EpiCare, providing a first useful benchmark for DTRs.

3. **Data Collection and Policy Choice:** We will provide a more detailed rationale for our choice of SMART as the behavior policy for data collection. Essentially this is due to it being more representative of real-world medical data collection practices than would alternatives such as collecting the data via an online RL agent. We also plan to include example outcomes for offline RL algorithms trained on data collected by an online RL behavior policy as recommended by two of the reviewers.

4. **Relationship to Real-world Data:** We will clarify that EpiCare is not intended to be fitted to or directly compared with real-world datasets. Instead, we'll emphasize its role as a generalized benchmark that captures key challenges common in longitudinal healthcare, without being tied to any specific disease. Fitting to real data would compromise EpiCare's generality, reducing its utility as a benchmark for RL in longitudinal care. Currently, OPE Is the main way in which the performance of RL algorithm performance is evaluated for real-world medical data. Our work provides the first medically relevant benchmark of OPE performance.

We are deeply grateful for the positive feedback and encouragement we've received from the reviewers. Your recognition of EpiCare's potential impact on the field of reinforcement learning in healthcare is truly motivating.
We appreciate the acknowledgment of:

- The critical need for a standardized benchmark in Dynamic Treatment Regimes (DTRs)
- Our efforts to capture essential complexities of healthcare decision-making
- The potential of EpiCare to drive meaningful progress in applying RL to healthcare
- The flexibility and scalability of our approach

Your positive comments validate the importance of our work and encourage us to further refine and expand EpiCare. We are committed to addressing the areas for improvement you've identified to make EpiCare an even more valuable tool for the research community.

### List of Major Changes

- Clarify the nature of EpiCare as a benchmark rather than a disease-specific simulator throughout the paper (especially the introduction, related work, and conclusion sections).
- Add a section providing clear guidance on how to use and interpret EpiCare results in practice.
- Expand the discussion on the rationale for using the SMART policy for data collection.
- Update Table 8 in the appendix and move it to the main body of the text, adding data for the performance of policies trained using data collected using an online behavior policy.
- Include SMART as one of the baselines in the tables throughout the paper.
- Add performance tables at realistic data levels to highlight current limitations.
 - Incorporate comparisons with real-world datasets like MIMIC-IV to illustrate similarities in complexity between EpiCare and real data.
- Updated EpiCare to include individual patient variation and compared performance of CQL with and without this variation.

### Patient Specific Effects

Here we show the performance of CQL before and after including patient-specific modifiers unique to each individual patient for Env 1 (see Q5&6 in the rebuttal for E79g)

| Environment | Original CQL Mean ± SE | Patient-Specific Mean ± SE |
|-------------|-------------------------|-----------------------------|
| Env 1       | 78.0 ± 0.9              | 65.20 ± 0.74                |
| Env 2       | 77.8 ± 0.3              | 62.13 ± 0.79                |
| Env 3       | 75.6 ± 0.8              | 55.89 ± 0.91                |
| Env 4       | 78.8 ± 1.2              | 59.37 ± 0.75                |
| Env 5       | 78.2 ± 1.0              | 60.17 ± 0.77                |
| Env 6       | 80.0 ± 0.5              | 61.33 ± 0.77                |
| Env 7       | 77.9 ± 0.8              | 61.68 ± 0.76                |
| Env 8       | 76.6 ± 1.6              | 58.23 ± 0.77                |
| **Mean**    | **77.36 ± 1.07**        | **60.50 ± 0.78**            |

---

### Decision · Program_Chairs · 2024-09-26

**Decision:**

Accept (Poster)

**Comment:**

It seems like all reviewers agree that this paper would make a meaningful contribution to the set of available resources for assessing RL performance on dynamic treatment regimes in the medical setting.

Some repeated concerns include (1) the clarification of the utility of this simulated environment  as a benchmark, and (2) justification of specific simulation choices (eg. the SMART policy for data collection). It seems like this feedback is most critical for the authors to reflect on in future iterations of this work -- how does this simulation setting and the modifiable parameters compare to real world settings? A discussion of related case studies or some reflection on the validity of the data design and data collection process would be helpful for the reader to better understand the rationale behind such choices.

Overall, despite these limitations, this work has clear value: RL benchmarks tend to operate in more abstract simulation settings, and there's much value in introducing benchmarks that capture more realistic properties of the type of data encountered in real world settings, especially in the medical domain. This is a meaningful intervention in that direction.